# Anti-correlated feature selection prevents false discovery of subpopulations in scRNAseq

Scott R. Tyler [1,2] ✉, Daniel Lozano-Ojalvo [3], Ernesto Guccione [2,4,5] & Eric E. Schadt [1] ✉

While sub-clustering cell-populations has become popular in single cell-omics, negative controls for this process are lacking. Popular feature-selection/clustering algorithms fail the null-dataset problem, allowing erroneous subdivisions of homogenous clusters until nearly each cell is called its own cluster. Using real and synthetic datasets, we find that anti-correlated gene selection reduces or eliminates erroneous subdivisions, increases marker-gene selection efficacy, and efficiently scales to millions of cells.

A frequent first task in performing cell-type identification from scRNAseq is feature selection to identify genes that add structure to the dataset based on various statistical properties, prior to unsupervised clustering. These algorithms differ from feature-selection applied in the context of a classifier for cell-type label transfer[1]. Current approaches to feature selection prior to unsupervised clustering in single cell -omics include measures of the relationship between a gene's mean and variance (i.e., overdispersion)[2–4], a gene's mean and dropout rate[5], a gene's deviance from an expected distribution[6,7], or degree of zero-inflation[7]. Conceptually, these algorithms operate based on examining each individual gene's expression distribution, assessing its statistical properties relative to an assumed distribution, ranking genes by their deviation from this expected distribution. An open problem however is how algorithms handle the "null-dataset;" that is, when there is only a single cell-identity present.

Given the popularity of sub-clustering (i.e., iteratively subdividing the initially identified clusters)[8–11], it is important to know that these groups are not being erroneously subdivided, thus producing false subtypes[12]. While novel sub-populations of interest should always be validated via bench-biology methods, an algorithmic assurance that one is not being misled can save money and years of effort attempting to validate erroneously discovered "novel sub-populations." Given the imperfections in clustering algorithms[13], sub-clustering itself can be a valid practice, because a single round of clustering may be insufficient to fully divide a dataset into its constituent groups. However, we must

have confidence that such algorithms will correctly identify single populations, preventing the false discovery of nonexistent subpopulations. In the case of a single cell population, either (1) a feature selection algorithm would accurately report that there are no genes that define sub-populations, or (2) the clustering algorithm would determine that only a single cluster is present.

In this work, we develop a new feature selection algorithm that (1) can effectively identify when a single population is present, through correctly identifying no sub-population specific genes within the dataset, (2) is highly sensitive to true signal that distinguishes subpopulations, and (3) can scale to 1-million cells, enabling discovery of subtly different populations.

## Results

### The anti-correlation-based feature selection algorithm

We sought to devise an algorithm to identify cell-type marker-genes that would both identify subpopulations of cell-types with high accuracy, and simultaneously solve the null-dataset problem. We thus began from first principles, asking the question: "what is a cell-type?". Traditional molecular biology has defined cell-types based on distinct cellular functions that are concordant with expression of distinct sets of genes: "marker-genes" (Fig. 1a), that often include hierarchical mutually exclusive gene expression. For example, in the pancreas the gene *NEUROD1* is a pan-endocrine marker, expressed in many different cell-types but should be mutually

[1]Department of Genetics and Genomic Sciences, Icahn School of Medicine at Mount Sinai, New York, NY, USA. [2]Department of Oncological Sciences, Tisch Cancer Institute, Icahn School of Medicine at Mount Sinai, New York, NY, USA. [3]Department of Dermatology, Icahn School of Medicine at Mount Sinai, New York, NY, USA. [4]Center for Therapeutics Discovery, Department of Oncological Sciences and Pharmacological Sciences, Tisch Cancer Institute, Icahn School of Medicine at Mount Sinai, New York, NY, USA. [5]Bioinformatics for Next Generation Sequencing (BiNGS) Shared Resource Facility, Icahn School of Medicine at Mount Sinai, New York, NY, USA. ✉e-mail: scottyler89@gmail.com; eric.schadt@mssm.edu

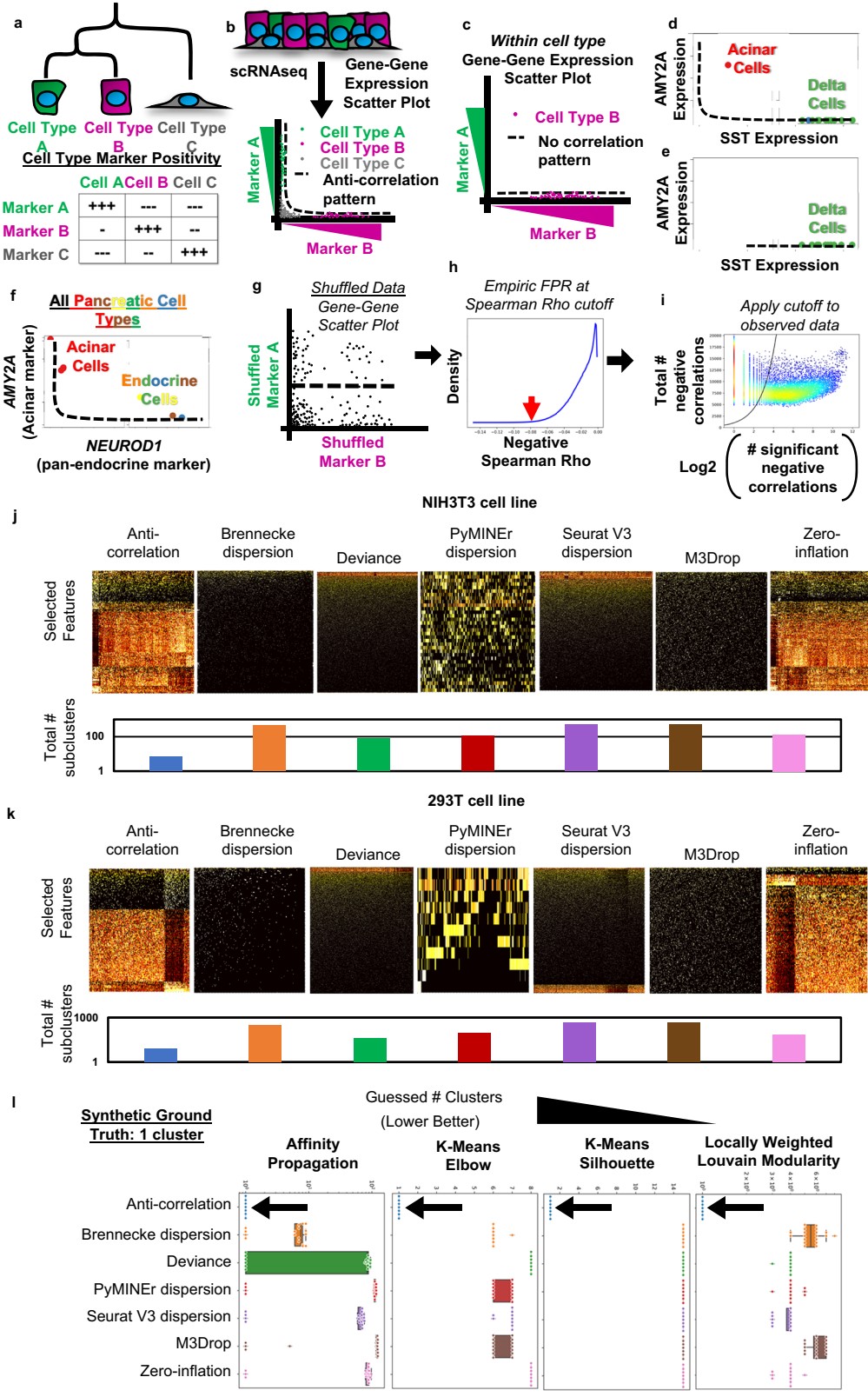

exclusively expressed from exocrine marker-genes[14]. If we accept this definition of cell-type and -lineage specific genes, we can algorithmically discover marker-genes from scRNAseq, as these genes will show a statistical excess of negative correlations with other genes (Fig. 1b). Given this premise, if only a single cell-identity is present in a dataset, we would expect an absence of an anti-correlation pattern since the cells of other cell-identities would not

be present (Fig. 1c). Indeed, looking at known marker-genes from different cell types in the pancreas (i.e., *AMY2A* expressed in acinar cells and *SST* expressed in delta cells), we see the expected anti-correlation pattern between *AMY2A* and *SST* (Fig. 1d), which disappears when examining subsets comprised of only a single cell type (Fig. 1e). Notably, the anti-correlation pattern holds for lineage-markers as well as cell-type markers (Fig. 1f).

**Fig. 1 | Anti-correlation algorithm premise and passage of the null-dataset problem. a** The logical schematic behind anti-correlation-based feature selection. **b** As a scatter plot where expression of marker A is plotted against marker B, cells of type A and B will form an L-shaped anti-correlation pattern, while cell-type C would express low levels of both marker A and B. **c** This anti-correlation pattern would disappear when examining a single population of cells. **d** The anti-correlation pattern of marker-genes appears in an example dataset[3], where high expression of *AMY2A* in acinar cells forms an anti-correlation pattern with *SST* in delta cells of the pancreas. **e** The anti-correlation pattern between *AMY2A* and *SST* disappears when only subset for delta cells. **f** The anti-correlation pattern is also present in lineage-marking-genes as shown by the pattern of *AMYA2* and *NEUROD1*, which labels all endocrine cells of the pancreas. **g** The anti-correlation-based feature selection algorithm first calculates a null background of Spearman correlations based on bootstrap shuffled gene-gene pairs to calculate a background. **h** Next the cutoff value closest matching the desired false positive rate (FPR) is determined. Displayed is a histogram of the bootstrap shuffled null-background of Spearman correlations less than zero. **i** Lastly genes which show more significant negative correlations (x-axis) than expected by chance (black line), given the gene's number of total negative correlations (y-axis), are selected: i.e., those to the right of the cutoff line. These are then used to calculate the False Discovery Rate (FDR) for each gene (See Methods for details). Heatmaps of selected features, and the total number of subclusters for each method of feature selection paired with AP clustering, when algorithms were allowed to sub-divide iteratively for homeostatic cell line scRNAseq: (**j**) NIH3T3, (**k**) HEK293T. **l** Boxplots indicating the total number of clusters identified by each method of feature selection (box colors) and clustering (noted in panels). Boxplots show lines that extend to minimum and maximum, with the box bounds from 25th to 75th percentile, and center denoting the median (*n* = 20). Source data are provided as a Source Data file.

Using these observations, we constructed an algorithm that identifies genes with an excess of negative correlations relative to what would be expected if the gene were un-patterned, as empirically measured with a bootstrap shuffled null background (Fig. 1g, h). We then select genes that have an excess of negative correlations, controlling for false positives (FPs) by setting an appropriate false discovery rate (FDR) (Fig. 1i). Overall, this procedure selects the genes that have significantly more negative correlations with other genes than would be expected by chance (See Methods for details). While others have performed small-scale experiments using positive correlations for feature selection, it was deemed infeasible due to computational run-time[15]; here we create an open-source, efficient implementation in python to overcome this barrier, but focus attention on negative correlation patterns as opposed to positive. Additionally, our implementation differs from standard methods in principle; here we identify the information held between genes, in contrast to the standard approach of measuring single-gene conformity to an expected distribution or pattern. Additionally, future feature selection algorithms may build on the concept of cross-feature measures (such as measuring cross-feature information or entropy); however, here we focus on the use of cross-feature correlations.

Given our reasoning that the anti-correlation pattern should go away when examining data representing only a single cell-type (Fig. 1c), with preliminary support for our rationale in a single dataset (Fig. 1e), we hypothesized that anti-correlation-based feature selection would be sufficient to solve the null-dataset problem, while status quo algorithms may not adequately solve this problem. With the null-dataset, no "cell-type or cell-state specific genes" should be identified as this is a single population of cells. We tested this hypothesis by performing feature selection and affinity propagation (AP)-based clustering on two datasets composed of scRNAseq from homeostatic cell line culture from NIH3T3 (Fig. 1j) or HEK293T cells (Fig. 1k), which we anticipate would capture the biologically relevant variation in only a single clustering round, and any attempt to *further* subdivide beyond that should be algorithmically blocked. Indeed, the anti-correlation algorithm allowed for only a single round of clustering, while the other algorithms tested allowed for further subdivisions (Fig. 1j, k).

**Efficacy on the null- and recursion-to-completion problems**

While this preliminary evidence suggests that anti-correlation-based feature selection solves the issue of FPs from sub-clustering homogenous populations, real-world datasets do not harbor a "ground-truth." We therefore simulated a single cluster using Splatter which produces negative binomially distributed gene expression matrices[16]. We performed feature selection using the noted algorithms[2–7] and passed these features to four different clustering algorithms including Affinity Propagation, K-means + Elbow-rule, K-means + Silhouette, and locally weighted Louvain modularity (See methods for algorithm details). In all cases, the anti-correlation-based method for feature selection detected no valid features within a single population of cells, thus addressing the null-dataset problem, while all other feature selection and clustering algorithm combinations failed the null-dataset problem, selecting noisy features that resulted in at least several clusters (Fig. 1l). Note that most feature selection algorithms frequently require the user to manually set the number of "discoveries" or selected features, which is likely a key contributor to this failure of the null-dataset problem when using standard feature selection approaches.

Without an algorithmic check to prevent erroneous sub-clustering, one could recursively divide a dataset until it is fully subdivided (each individual cell representing its own cluster), here dubbed "recursion-to-completion" (Fig. 2a). In practice, this would indicate that someone analyzing a scRNAseq dataset could always decide to sub-cluster a "cluster of interest" and report a "novel subpopulation" of cells, resulting in false discoveries. We created a pipeline to assess the robustness of each feature selection algorithm to the recursion-to-completion problem, by recursively performing feature selection and clustering on each progressive subdivision, moving from detecting global structure, towards increasingly local structure (Fig. 2b). This repeated process of feature selection holds three benefits over maintaining the original features with altered cluster resolution (1) It allows use to use an 'empty' feature list as an indication that no more clusters exist (the same method we used for passing the null-test), (2) allows for dynamic detection and subsequent refinement of compound correlation structures that differ at the global and local scale, and (3) does not incorporate noise from features enriched in unrelated lineages. For example, sub-clustering of T-cells, would not be enhanced by continued inclusion of epithelial cell markers from a prior round of clustering. Using repeated feature selection at each round of sub-clustering should enhance the ability to identify additional, previously unexplored sub-types. However, as with the recursion-to-completion task, at some point, an algorithm should halt and prevent further sub-division when no additional meaningful structure exists.

We applied this pipeline (Fig. 2b), assessing performance on the recursion-to-completion task with four publicly available datasets from differing species and platforms including droplet-based UMI approaches (Fig. 2c) and full-length transcript single-cell and -nucleus RNAseq (sNucSeq) (Fig. 2d)[17]. Again, we found that standard feature selection methods enabled recursion-to-completion, often finding hundreds to thousands of clusters, while anti-correlation-based feature selection were robust to this problem (fewer overall clusters: $P \le 1e-4$ for TukeyHSD post-hocs; Fig. 2d, e). This demonstrates that anti-correlation-based feature selection is robust to differing technologies, species, and sequencing type, retaining the ability to minimize false sub-divisions.

To verify these results with known ground-truth, we simulated 4 clusters, and allowed each algorithm to iteratively sub-cluster until either no features were returned, or only a single cluster was identified.

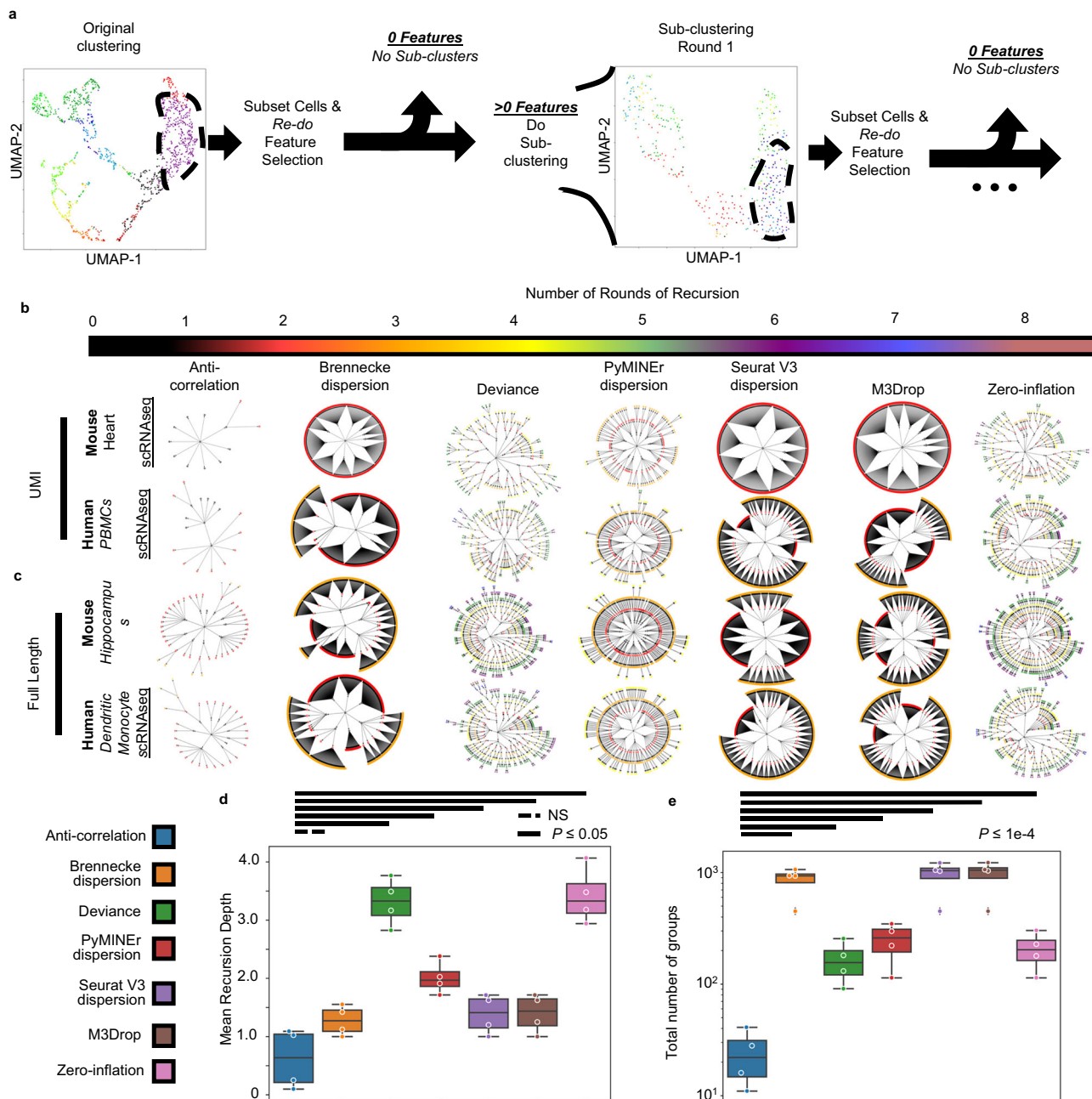

**Fig. 2 | Recursion-to-completion in real datasets. a** A schematic of sub-clustering is shown in the form of UMAP projections of the original dataset (left panel), and a sub-clustering iteration of a population found in the first round of feature selection and clustering (right panel). **b, c** In real datasets of varying technologies, status quo algorithms fail the recursion-to-completion problem while the anti-correlation-based approach prevented recursion-to-completion. Recursive clustering plots where each point indicates a cluster at a given recursive clustering recursion-depth as denoted in successive rings and color. **d** Boxplots of the mean recursion depth for each of the final sub-clusters for each noted method (1-way ANOVA with 2-sided TukeyHSD poshoc). **e** Boxplots of the total number of groups obtained through iterative sub-clustering (1-way ANOVA with 2-sided TukeyHSD poshoc). Boxplots show lines that extend to minimum and maximum, with the box bounds from 25th to 75th percentile, and center denoting the median. (**d, e:** $n = 4$ datasets). Exact $p$-values for all pairwise comparisons are availabe in Source Data file. Source data are provided as a Source Data file.

Consistent with our findings from real-world datasets, anti-correlation-based feature selection protected against erroneous sub-clustering, while other approaches allowed for several rounds of recursive sub-clustering, yielding hundreds to thousands of final 'clusters' (fewer average rounds of sub-clustering: $P = 4.5\text{e-}11$, $F = 114.1$, main-effects 1-way ANOVA; $P \leq 1.5\text{e-}5$ for TukeyHSD post-hocs; fewer total clusters: $P = 4.6\text{e-}13$, $F = 222.2$, main-effects 1-way ANOVA; $P \leq 3\text{e-}11$ for TukeyHSD post-hocs); Supplementary Fig. 1a. These simulated data demonstrate that anti-correlated feature selection guards against

erroneously splitting a single population of cells, while the algorithms tested here enable false discoveries of what appear to be "novel sub-types."

**Feature-selection accuracy in simulation and real-world data**
We next sought to determine the overall accuracy of these feature selection algorithms, where ground-truth differentially expressed genes (DEGs) should be selected by feature selection algorithms, and non-DEGs should not be selected. To this end, we used Splatter to

simulate datasets comprised of 4, 6, 8, and 10 clusters. Our anti-correlation algorithm had the best accuracy, Mathew's Correlation Coefficient (MCC), False Positive Rate (FPR), FDR, precision, true negative rate (TNR) compared to other feature selection algorithms (Supplementary Fig. 1b). However, anti-correlation-based feature selection had average recall (also called sensitivity or false negative rate (FNR)); this is explained however, by Splatter's wide-spread co-expression of *all* genes in *all* clusters (Supplementary Fig. 2a). In other words, using Splatter, *all clusters* express the "marker-genes" of *all other* clusters, therefore blunting the anti-correlations of marker-genes seen in practice (Fig. 1), thus reducing the apparent sensitivity. SERGIO however is a gene regulatory network (GRN) based scRNAseq simulation approach that more accurately represents empirical scRNAseq datasets[18] and does not induce co-expression of all marker genes in all clusters (Supplementary Fig. 2b). Using this simulation paradigm anti-correlation-based feature selection outperformed other approaches by every metric including recall/sensitivity, with the exception of deviance and zero-inflation, which achieved higher recall through selecting nearly all genes as shown by a FPR near 1.0, likely as an effect of Splatter and SERGIO simulations not matching distributional assumptions of these approaches (Supplementary Fig. 1c).

Using these simulations, we further performed a hyper-parameter sweep to identify effective values, and the trade-off in sensitivity and false-positives, among a total of 11 machine learning metrics for classification problems. We found the anti-correlation algorithm was very effective at preventing FPs among nearly all functional hyperparameters (Supplementary Fig. 3a). Where there was not great variability in performance with SERGIO under different FPR and FDR hyperparameter choices, Splatter showed a trade-off, but was still extremely robust with FPR = 0.001 and FDR = 0.066–0.25 hyperparameter settings (defaults: FPR = 0.001 and FDR = 0.066).

Additionally, we sought to clarify which hyperparameters were necessary to pass the null-test. Given the 'detection of a single population' is done indirectly, through returning zero selected features, we benchmarked the anti-correlation-based feature selection algorithm across these hyperparameters to identify the number of genes selected, and whether the algorithm passes the null-test. We observed that no features were selected (Supplementary Fig. 3b) in our null simulations, uniformly passing the null-test (Supplementary Fig. 3c), identifying no clusters whenever the FDR parameter was set to values ≤ 0.5. This result demonstrates strong robustness (good specificity) across hyperparameter space, ensuring that when following our guidance, one will be protected from false discoveries.

Feature selection prior to unsupervised clustering is intended to identify the cluster specific signal within the noise; we therefore sought to quantify this directly. Re-processing of previously published datasets, comparing results with previously published cluster labels often yields only ~50–70% concordant label results[19]; however, simulations provide a known ground-truth. We therefore benchmarked cluster results, with differing levels of signal to noise within the simulations, with and without our feature selection method paired with 8 analysis pipelines. This included our implementation of locally weighted Louvain, Scanpy's Louvain, Scanpy's Leiden, Sc3s, Seurat's Louvain, scCAN, scDHA, and SINCERA[3,20–25]. Indeed, with both SERGIO and Splatter simulations, the use of anti-correlation-based feature selection increased clustering accuracy by selecting signal from the noise when signal was sufficiently low in the original dataset (Supplementary Fig. 4). We further observed that our implementation of locally weighted Louvain modularity (See Methods) showed the greatest clustering efficacy followed by Seurat's Louvain[21], and scCAN[23].

To demonstrate efficacy with real-world datasets, we used seven pancreatic datasets[3,26–30], and found that the anti-correlated genes were either tied for, or had significantly higher *p*-value significance rank and precision for pancreatic specific genes based on gProfiler/

Human Protein Atlas tissue enrichment compared to other algorithms (Supplementary Fig. 1d)[31,32].

## Anti-correlation scalability in 1-million cells

To assess the practical scalability of anti-correlation-based feature selection, we re-processed and ran a larger dataset (245,389 cells) from a *Tabula Muris* data-release[33]. The full feature selection process took 60.95 min, while calculating the cell-cell correlations, distance, and clustering were far more computationally intense taking several days (see Methods for clustering details) (Fig. 3a). These findings show that anti-correlation-based feature selection should not be a major limiting factor for large datasets.

We also sought to demonstrate our feature-selection approach's utility in safe sub-clustering in practice; to this end, we focused on a cluster whose marker genes included insulin/amylin (*Ins1/2*, *Iapp*) and glucagon (*Gcg*), the markers for pancreatic beta and alpha cells, respectively, indicating that this cluster was insufficiently divided in the first clustering round. We performed sub-clustering with anti-correlation, identifying leukocyte, alpha-, beta-, and delta-cell populations. We further sub-clustered the insulin high population, and unexpectedly found the rare[34] population of pancreatic-polypeptide (*Ppy/Pyy*) expressing PP-cells (Fig. 3b), a cluster comprising only 0.01% of the original dataset. Attempting to further sub-divide PP-cells yielded no usable features, thus showing that anti-correlation-based feature selection can facilitate extremely sensitive sub-clustering to identify rare biologically meaningful populations from large datasets, while also preventing errant subdivisions.

As seen in the final sub-cluster round, however, while anti-correlation-based feature-selection is biologically accurate and answers the question: "Should this cluster be sub-clustered?", it does not ensure that downstream algorithms will select the correct number of clusters; this remains an outstanding problem as previously reported[12]. However, passing the first step of successfully identifying a homogeneous population, through anti-correlation-based feature selection, provides confidence that meaningful structure existed in the parent population.

One real-world scenario that can be encountered, however, is the use of multiple batches or technologies simultaneously. If appropriate caution is not exercised, such situations can introduce Simpson's paradox, which can result if technical effects are layered on top of the biological effects, changing the overall global correlation structure as a result of mixing two different local correlation structures. Indeed, examining a single donor sample from the *Tabula Sapiens* dataset whose cells were assayed in parallel using two differing technologies (10X Genomics' Chromium and SMART-seq2), these technical replicates both showed negative correlations when assayed independently with respect to the two technologies employed. However, when analyzed jointly, these negative correlations became positive, giving rise to Simpson's paradox, despite the fact that these were technical replicates (Supplementary Fig. 5). While common defaults in pipelines will take the intersection of feature-selection runs performed on each individual batch, this will only capture the intersection of biological effects, therefore leaving users at risk of discarding batch confounded, but biologically meaningful variation. This highlights the need to perform feature selection analysis without confounded technical and biological signal.

Unlike the above situation however, multiplexing within a single technology provides a possible solution, because only a single batch effect is applied to all samples simultaneously. To assess performance under this paradigm, we used our anti-correlation algorithm with a dataset of 1-million PBMCs from 24 donors: 12 healthy, and 12 with type-I-diabetes (T1D). Our feature selection algorithm was again not a limiting factor, taking only 3.3 h (Fig. 3c), with calculating the cell-cell

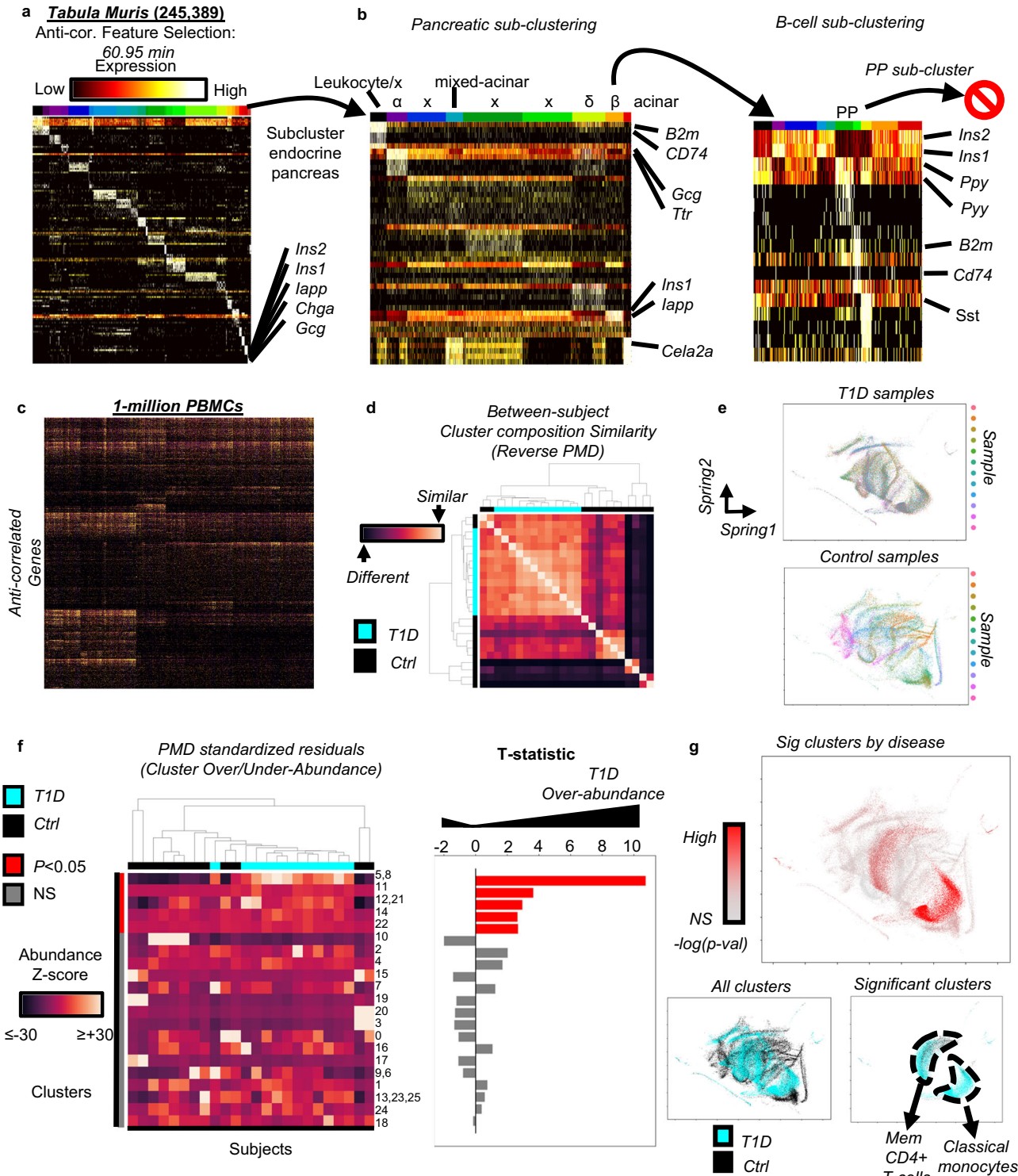

**a** _Tabula Muris_ **(245,389)** Anti-cor. Feature Selection: _60.95 min_ Expression — Low / High — Leukocyte/x, mixed-acinar, α, x, x, x, δ, β, acinar — Subcluster endocrine pancreas — _Ins2, Ins1, Iapp, Chga, Gcg_

**b** _Pancreatic sub-clustering_ — _B2m, CD74, Gcg, Ttr, Ins1, Iapp, Cela2a_ — _B-cell sub-clustering_ — PP — _PP sub-cluster_ — _Ins2, Ins1, Ppy, Pyy, B2m, Cd74, Sst_

**c** _1-million PBMCs_ — Anti-correlated Genes

**d** _Between-subject Cluster composition Similarity (Reverse PMD)_ — Similar / Different — T1D / Ctrl

**e** _T1D samples_ — Spring2 / Spring1 — Sample — _Control samples_ — Sample

**f** T1D / Ctrl — P<0.05 / NS — _PMD standardized residuals (Cluster Over/Under-Abundance)_ — Abundance Z-score ≤−30 / ≥+30 — Clusters — Subjects — **T-statistic** — _T1D Over-abundance_ — −2 0 2 4 6 8 10 — 5,8; 11; 12,21; 14; 22; 10; 2; 4; 15; 7; 19; 20; 3; 0; 16; 17; 9,6; 1; 13,23,25; 24; 18

**g** _Sig clusters by disease_ — High / NS −log(p-val) — _All clusters_ — T1D / Ctrl — _Significant clusters_ — Mem CD4+ T-cells — Classical monocytes

distance matrix, converting to k-Nearest-Neighbor graph, and clustering taking days. Note that the full dataset was used for all steps, rather than clustering on a subset and performing label transfer. In this way, we will have the greatest power to detect novel sub-populations. As previously mentioned, several instances of over-clustering occurred, which we manually re-combined where it was determined that UMI depth was the primary difference. Comparison to a previously published large PBMC dataset[35] enabled coarse-grained best-guess cluster labels (Supplementary Fig. 6, Supplementary Datasets 1–8). However manual analysis was still necessary for accurate labeling.

### Unsupervised discovery of subtly differing sub-populations

Among the B-cells, we observed low abundance *BLIMP1+/XBP1+* plasma cells (cluster-11)[36] (Supplementary Fig. 7), that contrasted with CD268+ mature B-cells (clusters-2, −15, −16). Cluster-15 was characterized by up-regulation of *NFKB1*. Clusters-2 was *CD23+*, while cluster-16 cells were CD23-, but also contained two regions that were characterized by activated *TACI+/CD80+/CD5-* and non-activated TACI-/CD80-/CD5+ phenotypes, which may be consistent with B1 (with B1a/B1b subtypes)[36]. However, definitive identification and further refinement of subsets, is an area of ongoing research[37], especially

**Fig. 3 | Application to large real-world datasets. a** A heatmap of the top 5 marker genes per cluster are shown for the primary lineages from the full senescent *Tabula Muris* dataset[33], with the last cluster representing a mixture of cell-types from the endocrine pancreas. **b** When subclustered with anti-correlated feature selection, mixed-cell-type droplets (x) as well as classically described leukocyte, α, δ, β, and acinar populations were discovered. Subclustering β cells discovered mixed-lineage droplets with δ and leukocyte cells as well as the rare PP-cell population, but additional subclustering of PP-cells was prevented by anti-correlation-based feature selection. **c** Selected features for clustering 1-million PBMCs. **d** Subject-level reverse Percent Maximum Difference (rPMD), shows that Type-1-Diabetes (T1D) subjects are more similar to each other, while control PBMCs are more diverse by cluster composition. **e** A spring embedding of a subset of cells from each cluster, color-coded by donor, with sub-plots for T1D and control subjects, showing large-

scale uniformity in T1D compared to the heterogeneous control samples. Note that this is for display purposes only, was not used in analysis, and does not represent cell-cell distances, but rather a display of the graph used for clustering. **f** A heatmap of PMD standardized residuals, which correspond to the significance of how different each subject's relative abundance of all clusters differs from the null expectation of no-difference between subjects. A matching bar-chart shows the T-statistic of cluster level significance for each cluster's differential over-under abundance shown in the heatmap, comparing T1D to controls. Bars are color-coded by significance ($P < 0.05$ after Benjamini-Hochberg). Exact *p*-values available in Source Data file. **g** The spring embedding of the kNN graph is color-coded by significance of differential abundance for each cluster, and additionally color-coded by T1D/control status, then again subset for only the significant clusters. Source data are provided as a Source Data file.

given sensitivity limitations and post-transcriptional regulation in RNA-only assays.

Among T-cells (Supplementary Fig. 8a, b), we identified a BLIMP1+ mixed CD4-memory population (cluster-12,21)[38], *PECAM1*+ naïve *CD4* (cluster-13,23,25)[39,40], and memory and naïve *CD8* + T-cells, based on the same markers (clusters-9,6 and −0, respectively) (Supplementary Fig. 8c, d), and *FOS*+/*CD97*+[41] early activated T-cells, which rapidly downregulate CD4/CD8 after activation[42] (cluster-17) (Supplementary Fig. 8f). Interestingly, a subset of CD4mem cells appeared to show high expression of the *TOX* exhaustion marker gene[43]. To further refine this population, which was also moderately increased in T1D (Fig. 3g; T-statistic = 2.95, $P = 2.2e-2$, BH-corrected *t* test; Supplementary Datasets 3, 4), we subset and re-performed feature selection and clustering, successfully identifying 3 discrete populations (Supplementary Fig. 9a, b; Supplementary Datasets 7), with the largest *CD4*+ effector cluster existing along a cell-identify marked by a *STAT4/RUNX2-to-TSHZ2/ICOS* expression continuum (Supplementary Fig. 9c)[44,45]. A remaining *CD4*+ memory subcluster was *CD25*+/*FOXP3*+/*TOX*+, which is consistent with Tregs[46] (subcluster-7). The final low-abundance cluster (subcluster-11) was characterized by *PECAM1* positivity indicating that these were likely incorrectly co-clustered naïve *CD4* + T-cells from the prior clustering-round (Supplementary Fig. 9).

In addition to simply characterizing the many different subsets of PBMCs, we further sought to investigate their disease relevance. Notably, we found that the T1D samples were highly similar to each other based on cluster composition, whereas the healthy controls were far more diverse as measured by reverse Percent Maximum Difference (PMD) (1-PMD, Fig. 3d, e; Supplementary Dataset 1). PMD and its standardized residuals quantifies subject-level similarity based on cluster composition robustly to differences in subject level sampling depth[47].

We found several significantly differentially over-abundant clusters, however, the most differentially abundant was a subset of *CD14*+/*CCR2*+/*CD115*+ classical monocytes (Supplementary Fig. 10a–c; T-statistic = 10.7, $P = 1.34e-25$, BH-corrected *t* test; Supplementary Datasets 3, 4)[48] appearing uniformly over-abundant in T1D subjects (cluster-5,8; Fig. 3f, g; Supplementary Datasets 1–8)[49–51]. When comparing this population to its most closely related monocyte population (cluster-10), we found that the T1D over-abundant cluster had significantly higher expression of the activation marker *FOSB*, and functional modulators *NAMPT*[52] and *HIF1A*, which is induced in inflammation, even in normoxic conditions[53,54] (Supplementary Fig. 10d); DEGs between these populations are available in Supplementary Datasets 8. However, further research will be needed to interrogate and independently confirm these findings.

## Discussion
Overall, our findings demonstrate that anti-correlation-based feature selection solves the null-dataset and recursion-to-completion problems, outperforms others in overall feature selection accuracy, and

works with both UMI and full-length sequencing methods. These properties can prevent wasted time and money for bench-practitioners attempting to validate novel sub-populations by providing an algorithmic check to false discoveries in scRNAseq, and can scale to the size of modern datasets. Lastly, our python package (titled anticor_features) is open-source, pip installable, and compatible with SCANPY/AnnData[25] to enable broad adoptability.

## Methods
### Example of anti-correlation principle on pancreatic dataset
A previously published scRNAseq dataset and annotations were used for scatter plots of *AMY2A* for acinar cells, *SST* for delta cells, and *NEUROD1* for endocrine cells (Fig. 1d–f)[3].

### Normalization of scRNAseq datasets to be used for benchmarking
Due to large variation (often orders of magnitude differences) in total UMI counts across cells and it's downstream effects on cell-to-cell distance metrics, we normalized each cell within UMI based datasets through bootstrapped UMI downsampling as described here: https://bitbucket.org/scottyler892/pyminer_norm. In brief, a cutoff is selected for both the number of observed genes in a cell as well as the number of total UMI observed in a cell. Cells not meeting these criteria are removed, and all other cells are normalized through UMI downsampling. UMI downsampling is done through simulating the transcriptome of a given cell, and randomly selecting N transcripts, where N is the desired number of total UMI for each cell to have, in this case 95% of the cutoff used for total UMI count. Thus, each cell is randomly sampled to the same UMI depth.

To normalize full-length sequencing datasets with TPM or similar units, we created a variant of quantile normalization we call truncated quantile normalization. First a cutoff ($g$) is selected for the number of genes to be expressed in each cell in the final normalized dataset. Next, cells with fewer than $g + 1$ genes expressed are removed, then for each cell, the transcriptome is subtracted by the expression value of gene $g + 1$ for that cell, thus setting the $g + 1$ gene's expression to zero, leaving the remaining top $g$ expressed genes with >0 expression in all cells. All negative values are then set to 0. For ties at the expression-level of $g$ that would result in differing number of observed genes, genes are randomly selected to be preserved or set to zero stochastically. This yields a vector for each cell for whom the top expressed $g$ genes are kept, but shifted downwards in a manner that does not introduce an artificially large gap between the lowest expressed gene ($g$) and zero. These top $g$ genes for each cell are then quantile normalized. This process is implemented in the pyminer_norm pip package, and can be called from the command-line on tsv files:

```
python3 -m pyminer_norm.quantile_normalize -i in_file.tsv -o out_file_qNorm.tsv -n 2000
```
to perform truncated quantile normalization on the top 2000 genes for each cell.

### NIH3T3 and HEK293T cell line datasets

This dataset was downloaded from 10x Genomics' website at (https://support.10xgenomics.com/single-cell-gene-expression/datasets/3.0.2/1k_hgmm_v3). The cells of mouse or human origin were separated into distinct datasets for our purposes here based on the sum of reads that mapped to each species' transcriptome, while doublets were excluded. In the case of both human and mouse references, cells were kept that had >3162 counts mapping to hg19 or mm10 for HEK293T and NIH3T3 respectively, cells were also only kept if they had >1000 genes observed. The remaining cells were then downsampled to 3003 counts for each dataset to normalize for variable count depth that otherwise spanned two orders of magnitude.

### Affinity propagation

Our implementation of affinity propagation was based on the sklearn sklearn.cluster.AffinityPropagation function, in which the preference vector is initialized to the row-wise minimum of the input matrix; in this case, the negative squared Euclidean distance of the Spearman correlations across all cells. We observed that as datasets scale, the original affinity propagation algorithm fragments single populations into many small populations that were similar to each other. We therefore follow the original affinity propagation results with an analysis that calculates the distance (in affinity space) between cluster centers (also called exemplars). The standard deviation of within-cluster affinities is then calculated. For each cluster-cluster pair from the original affinity propagation cluster results, we then determine the number of combined standard deviations required to traverse half the Euclidean distance in affinity space between two cluster centers. This measure is the number of standard deviations needed to reach the waypoint between two cluster centers. Because these are standard deviation measures, we can convert these to transition probabilities, as with a Z-score, using the scipy.stats.norm.sf function. This creates a cluster x cluster matrix of transition probabilities; this probability matrix is then subjected to dense weighted Louvain modularity. Final clusters are determined by the results of this procedure, where AP clusters that were determined by Louvain modularity to belong to the same community are merged. All code and cluster for the affinity propagation with merged procedure can be accessed through running PyMINEr with the appended arguments: " -ap_clust -ap_merge" at the command line or interactively via the pyminer.pyminer.pyminer_analysis function using the arguments: ap_clust=True, ap_merge=True.

### Clustering – K-means with Elbow and K-means with silhouette

First each dataset (already log transformed) was subset for the genes selected by the given feature selection algorithm, then genes were min-max linear normalized between 0 and 1. K-means clustering was performed using the sklearn.cluster KMeans function. For the elbow rule, the sum of squared Euclidean distances of samples to their cluster center was used in conjunction with the given k value. We took the elbow to be the value of k which yielded the minimum distance to the origin.

For the silhouette method, we calculated the average silhouette score with the sklearn.metrics silhouette_score function, and sample level silhouettes calculated with the silhouette_samples function. The number of clusters was selected by moving from k = 1 to k_max, testing for whether there existed a cluster whose maximum sample level silhouette was less than the average silhouette score for the whole dataset (as determined by the silhouette_score function).

### Clustering – Locally weighted Louvain modularity

We created a kNN graph embedding and subjected it to Louvain modularity as follows:

1. Calculate Spearman correlation of all cells against all other cells (matrix: **S**).
2. Calculate the inverse squared Euclidean distance matrix from the Spearman matrix (matrix: **D**), divided by the square-root of the number of cells. In this matrix, cells that are more similar to each have higher values, and cells that are dissimilar have lower values, inversely proportional to the squared Euclidean distance.
3. For each cell, $i$, (i.e.,: row in matrix **D**) subtract the upper 95th percentile (or top 200th closest cell, whichever yields fewer connections) of distance vector ($\mathbf{D}_i$), then mask all negative values to zero, thus creating a weighted local distance matrix (matrix: **L**).
4. To ensure that all cells are on an equivalent scale, each row in **L** is divided by it's maximum ($\mathbf{L}_i = \mathbf{L}_i / \max(\mathbf{L}_i)$).
5. The normalized local distance matrix **L** serves as the weighted adjacency matrix for building the network for weighted Louvain modularity.

The locally weighted adjacency matrix was subjected to Louvain modularity as implemented in the python pip package: python-louvain.

### Sc3s clustering

Sc3s requires user specified k selection (number of clusters), with k needing to be greater than or equal to 2[22]. The "elbow method" defined above was also used here to select the "best" clustering result among k over the full range between 2 and 15 possible clusters using the following call:

```
sc3s.tl.consensus(local_adata, n_clusters = [2,3,4,5,6,7,8,9,10,11,12,13,14,15])
```

### Scanpy's Leiden and unweighted Louvain

Following scanpy tutorials, we implemented the leiden clustering algorithm as follows:

```
sc.pp.neighbors(local_adata, n_neighbors = 10, n_pcs = 50)
sc.tl.leiden(local_adata)
```

or for Louvain modularity:

```
sc.pp.neighbors(local_adata, n_neighbors = 10, n_pcs = 50)
sc.tl.louvain(local_adata)
```

### Seurat's implementation of Louvain Modularity

Counts were scaled and normalized, followed by PCA reduction to either 50 PCs or the maximum number of valid computable components, depending on the dimensions of the matrix after feature selection.

```
sce <- Seurat::ScaleData(
            Seurat::NormalizeData(
            Seurat::CreateSeuratObject(counts)
        )
    )
sce <- Seurat::RunPCA(sce, features = features, npcs=n_pcs)
sce <- Seurat::FindNeighbors(sce, dims = 1:dim(sce@reductions[["pca"]])[2])
sce <- Seurat::FindClusters(sce)
cluster_labels <- unlist(Seurat::Idents(sce))
```

### scDHA and scCAN implementations

Data were log2(1+counts) transformed and analyzed by scDHA or scCAN as follows:

```
scDHA::scDHA(data)$cluster
scCAN::scCAN(data)$cluster
```

### SINCERA implementation

Following the SINCERA demo below:

(https://github.com/xu-lab/SINCERA/blob/master/demo/humanIPF.R)

We processed the data as suggested implementing a log transformation, Z-scores, followed by a PCA reduction, as with previous methods above:

```
data <- log2(1+sim@assays$data[['counts']][features,])
sc <- SINCERA::construct(exprmatrix=data.frame(data), sample
vector=colnames(sim))
sc <- SINCERA::normalization.zscore(sc, pergroup=FALSE)
sc <- SINCERA::setGenesForClustering(sc, value=features)
x <- SINCERA::getExpression(
  sc, scaled=T, genes=SINCERA::getGenesForClustering(sc))
      )
n_pcs < -min(50,dim(x)[1]-1)
pc_res < -prcomp(x)$rotation[,1:n_pcs]
print(dim(pc_res))
```

This was followed by k-means clustering with the gap statistic:

```
gapstats <- cluster::clusGap(pc_res,
          FUN = hclustForGap,
          K.max = 15,
          B = 3)
f < -gapstats$Tab[,"gap"]
SE.f < -gapstats$Tab[,"SE.sim"]
num_clust <-cluster::maxSE(f, SE.f)
cluster_res <- hclustForGap(t(x),k=num_clust)$cluster
```

## Implementation of other feature selection algorithms

Because each feature selection algorithm expects slightly different processing methods relative to each other (either normalized and log-transformed, or count data), we followed author guidance in implementation.

**PyMINEr's overdispersion pipeline.** is contained within the originally published full PyMINEr pipeline, but is also callable within python as follows:

```
feature_table = do_over_dispers_feat_select(ids = cell_ids,
                ID_list=gene_ids,
                in_mat = exprs)
```

**Seurat's overdispersion.** Per author guidelines, we log-normalized the input expression matrix and selected features as follows:

```
obj < - NormalizeData(CreateSeuratObject(exprs))
obj <- FindVariableFeatures(obj)
var_feat <- VariableFeatures(obj)
```

**Original Brennecke algorithm.** We used the implementation of the original overdispersion-based feature selection algorithm as implemented in the M3Drop package as follows:

```
Brennecke_HVG <- BrenneckeGetVariableGenes(exprs, fdr =
0.05, minBiolDisp = 0.5)
```

**M3Drop.** Unlike other most other feature selection algorithms, M3Drop allows for either a pre-specified FDR, or a pre-specified percentage of the transcriptome to select. In our testing using the FDR approach (which could theoretically solve that the null-dataset problem), we found that each dataset required fine tuning of this cutoff to provide reasonable results, and in the case of full-length transcript based approaches did not select any genes even in the full datasets, which are known to be biologically complex. We therefore sought a more realistic implementation that did not require manual tuning for each dataset, and therefore implemented the "percentage" approach within M3Drop so that a standard call yielded meaningful results regardless of dataset, without necessitating a manual inspection for hyperparameter selection for all datasets, which could also be seen as tuning hyperparameters to fit our expectations of the data. The implementation was as follows:

```
results      <-      M3DropGetExtremes(exprs,    percent = 0.05,
suppress.plot = TRUE)
```

Using the genes within the results$right section as the genes with an excess of zeros for the final selected genes.

## Deviance implementation

We implemented the deviance test in the HIPPO R package, using the default deviance cutoff of 50:

```
subdf = preprocess_heterogeneous(counts)
features = hippo_select_features(subdf, "deviance", 2, 50)
```

## Zero-inflation test

We implemented the zero-inflation test in the HIPPO R package, using the default zero-inflation cutoff of 2:

```
subdf = preprocess_heterogeneous(counts)
features = hippo_select_features(subdf, "zero_inflation", 2, 50)
```

## Details of anti-correlation feature selection algorithm

We aimed to develop an algorithm that identifies genes that have "too many" negative correlations below a dynamically selected cutoff that make the selected genes more negatively correlated with other genes than one would expect from random chance. To this end we began with a FPR of 0.001, for identifying a cutoff at which correlations should be counted as a "discovery" (D, where more significant), or "non-discovery" (ND, where less significant). Using a bootstrap shuffled null background, in which all discoveries (D) are false, because true positives (TP) are known to be equal to zero:

$$FP + TP = N(D) \qquad (1)$$

Where D is all discoveries, more significant that the cutoff. Therefore because this is measured from a bootstrap shuffled null background (i.e.,: TP = 0):

$$FP = N(D) \qquad (2)$$

Using this knowledge, we created the null background of gene-gene Spearman correlations is generated through randomly sampling 5000 genes, shuffling within-genes, such that a gene-gene correlation plot would have its x-y pairing shuffled, calculating pairwise Spearman correlations.

**Definitions. $E_o$:** the original expression matrix

*rand*: an integer vector of the length 5000 for the random samples within the space of 1..n, where n is the number of genes

$E_r$: The random subset matrix that is permuted as defined below: For i..$N(rand)$:

$$E_{r,i} = permute(E_{o,rand[i]})$$

Where $E_r$ provides a N(cell) x N(rand) matrix, which is a within-gene bootstrap shuffled version of a subset of the transcriptome, therefore unpairing the gene-gene pairs for measuring the null background of Spearman correlations.

In our testing, using a greater number of randomly selected genes, $N(rand)$, for the permutation based null-background did alter the null-distributions, as these distributions were stable at this sampling depth, and did not notably change the selected cutoffs. Note that the method of rank transformation for Spearman correlation effects the outcome; here we perform dense-rank transformation. Non-dense rank transformations frequently result in large gaps within the distributions because of ties. This is particularly important with count-based datasets where ties are frequent.

The null Spearman background matrix (**B**) was the symmetric $5000 \times 5000$ comparison of this sample (5000 choose 2 combinations).

For $i = 1..N(rand)$ and $j = 1..N(rand)$:

$$\mathbf{B}_{i,j} = Spearman\left(\mathbf{E}_{r,i}, \mathbf{E}_{r,j}\right)$$

Next, this **B** background matrix, of null Spearman rho values, is filtered for only values $\mathbf{B}_{i,j} < 0$, thus creating a negative correlation null-background; this is needed because the null background for values $\mathbf{B}_{i,j} > 0$ and values $\mathbf{B}_{i,j} < 0$ follow different distributions (Supplementary Fig. 2c), indicating the necessity to measure them independently. Self-comparisons and duplicate comparisons were also removed.

For $i = 1..N(rand)$, and $j = i + 1..N(rand)$:

$$\boldsymbol{b} = \left(\mathbf{B}_{i,j} \in \mathbf{B} \middle| \mathbf{B}_{i,j} < 0 | i > j\right)$$

Conceptually, this filtering is also important because the estimated number of FP for a given gene $i$ is dependent on the number of genes that are actually randomly distributed, or truly correlated. For example, gene X is co-regulated within a module of 2000 genes, while gene Y is not genuinely correlated with any other genes. Given that the number of genes is static and zero sum, this true positive co-regulation removes those genes from possible FP negatively correlated genes.

This null background *vector* (*b*) is used to calculate an the cutoff ($\boldsymbol{C_{neg}}$) that most closely matches the desired FPR (default = 1 in 1000 FPs), with a discovery considered as a Spearman rho value $< \boldsymbol{C_{neg}}$ in the gene-gene correlation matrix (**S**) calculated from the unshuffled original expression matrix ($\mathbf{E}_o$), This cutoff is used for the estimated FDR for the original intact unshuffled dataset.

Given that:

$$FPR = \frac{FP}{FP + TN} \tag{3}$$

and

$$N(b) = FP + TN \tag{4}$$

Because TP = FN = 0, given that *b* was generated from a bootstrap shuffled null. We therefore find that:

$$FPR = \frac{FP}{N(b)} \tag{5}$$

and therefore

$$N(b) * FPR = FP \tag{6}$$

Therefore, to identify the appropriate cutoff ($\boldsymbol{C_{neg}}$), that yields the FPR( = 1e-3 by default), we simply take the Spearman rho value of *b* that is located within the sorted background vector that gives the ratio of FPs to true negatives.

$$b_{sort} = sort(b) \tag{7}$$

Such that for $i = 1..N(b_{sort}) - 1$, $b_{sort,i} < b_{sort,i+1}$

We then calculate the $\boldsymbol{C_{neg}}$ cutoff, but taking the value at the index that gives the expected ratio of FPs to true negatives as determined by the FPR hyperparameter (default = 1e-3)

$$C_{neg} = b_{sort, \lfloor FP \rfloor} \tag{8}$$

Next, we use this empirically determined cutoff ($\boldsymbol{C_{neg}}$), applying it to classify "discoveries" of negative correlations in the correlation matrix **S** as calculated from the original, non-shuffled dataset ($\mathbf{E}_o$).

Where a discovery is defined as a Spearman rho value $\mathbf{S}_{i,j}$ less than the $\boldsymbol{C_{neg}}$ cutoff.

Again it is important to note two things: (1) the null distribution of Spearman correlations, are in fact two separate distributions concatenated around zero, for the null distribution of rho values < 0, and the null distribution of rho values > 0 (Supplementary Fig. 2c); and (2) that variable abundance of True Positives within the positive correlation domain will decrease the total number of comparisons that fall within the negative correlation domain of these distributions; these two distributions are therefore in competition with one another, meaning that they must be quantified independently. For these reasons, when applying the empirically measured cutoff ($\boldsymbol{C_{neg}}$) from the shuffled transcriptome, we must apply it only to the correlations falling below zero. To apply this cutoff ($\boldsymbol{C_{neg}}$) to the original expression matrix ($\mathbf{E}_o$), we first calculate the symmetric gene-gene Spearman rho matrix (**S**).

Next, the number of total ($T$) Spearman rhos values < 0 within **S** is tabulated for the application of our cutoff ($\boldsymbol{C_{neg}}$):

$$T_i = N\left(\mathbf{S}_{i,j} \in \mathbf{S} | \mathbf{S}_{i,j} < 0\right) \tag{9}$$

For $i = 1..n$, where n is the number of genes.

Note also, that $T_i$ sums to the total number of discoveries (D) and non-discoveries (ND).

$$T_i = N(D_i) + N(ND_i) = TP + TN + FP + FN \tag{10}$$

Where:

$$N(D_i) = N\left(\mathbf{S}_{i,j} \in \mathbf{S} | \mathbf{S}_{i,j} < \boldsymbol{C_{neg}}\right) = FP + TP \tag{11}$$

and

$$N(ND_i) = N\left(\mathbf{S}_{i,j} \in \mathbf{S} \middle| \mathbf{S}_{i,j} < 0 \middle| \mathbf{S}_{i,j} > \boldsymbol{C_{neg}}\right) = FN + TN \tag{12}$$

Further, the discoveries are comprised of both FP and true positives (TP), however, *which* individual values within the discovery class is a FP or TP is unknown. Using the FPR however, we can estimate the number of *expected* FPs given the total number of comparisons <0 for the given gene ($T_i$). In other words, if this gene were random in its negative correlations, then only a specific number of FPs would be expected ($\hat{FP}_i$), using $\boldsymbol{C_{neg}}$ as a cutoff.

$$\widehat{FP_i} = T_i * FPR \tag{13}$$

Therefore, with *FDR* defined as:

$$FDR = \frac{FP}{(FP + TP)} \tag{14}$$

We can estimate the *FDR* for each gene, determining if it has an over abundance of negative correlations compared to what is expected from the null distribution:

$$\widehat{FDR_i} = \frac{\widehat{FP_i}}{N(D_i)} \tag{15}$$

We then select genes that have a > 15x excess in discoveries relative the expected number of FPs under the null distribution assumption. This corresponds to an estimated $\widehat{FDR} = 0.066$ (1/15). This yields the set of all excessively negatively correlated genes (A):

$$A = \{gene_i \in genes | \widehat{FDR_i} < 0.066\} \tag{16}$$

Lastly, given that spurious positivity is still possible and even expected, we add one last layer of protection against false discoveries.

The positive/negative status of a single gene likely does not define a truly "novel subtype" – particularly in a technique such as single-cell -omics where stochastic dropout from random sampling is expected. We therefore apply an additional filter from the premise that the genes whose expression patterns separate meaningful populations should also be positively correlated with other genes that are following similar regulatory patterns. To select this population of genes, we find genes that have greater than 10 positive correlations above the positive correlation cutoff ($C_{pos}$), as calculated similarly to ($C_{neg}$) as described above.

$$M = \{gene_i \in genes | N(S_{i,j} | S_{i,j} > C_{pos} | i \neq j) > 10\} \qquad (17)$$

The final included features are the intersect of A and M:

$$F = A \cap M \qquad (18)$$

Overall, this means that genes must contain both an excess of negative correlations, and be a member of a "module" of at least 10 genes that move in concert.

### Recursion benchmarks

An initial run of locally weighted Louvain modularity was performed, then the given dataset was subset to contain only the cells of a given cluster in the prior round of clustering. Next, feature selection and locally weighted Louvain modularity was applied again, recursively until either each cell was called its own "cell-type"/cluster or produced "cell-types"/clusters with ≤5 cells.

Circular recursion graphs were displayed using networkx[55], with layout determined by the graphviz_layout(prog = 'twopi') layout[56].

### In silico recursive clustering benchmark

Four clusters were simulated using Splatter[16], and all algorithms were allowed to recursively select features, which were then subjected to locally weighted Louvain modularity until one of the following conditions were met: no features were selected, the clustering algorithm only found a single cluster, or the results of clustering formed groups of 5 or fewer cells.

### Real-world recursive clustering benchmark

The above described recursion procedure was applied to the previously released mouse heart scRNAseq dataset[57], and human PBMC dataset[58] for UMI based technologies, and mouse hippocampus single nucleus RNAseq[17] and human dendritic cell/monocyte[59] datasets were used for full length transcript sequencing based approaches. Each dataset was normalized as described above and is available in the repository site containing this benchmark: https://bitbucket.org/scottyler892/anti_correlation_vs_overdispersion in the data folder. The same recursive clustering procedure was followed as described for the in silico recursion benchmark above.

### Feature selection accuracy based on Splatter and Sergio simulations

For both simulation paradigms, we simulated 4, 6, 8, and 10 clusters. 2500 cells were simulated with 10,000 genes, of which 2000 were intended to be differentially regulated across clusters. Once simulations were completed, the datasets were downsampled down to 95% of the cell with the lowest total counts in the given dataset, using the pyminer_norm python package[47].

Splatter simulations were generated using the bin/simulate_data.R with the above described clusters, cells, and gene parameters. SERGIO simulations were generated from the bin/generate_sergio_sim.py script, which was called from the bin/simulate_data.R file. For each cluster, a single "master-regulator" gene was used to induce high expression of its child nodes in the GRN. The non-differentially

regulated genes were random negative binomial distributions added to the network with the np.random.negative_binomial function.

Similar to performing pathway analyses, a proper background list of genes is necessary for quantifying enrichment. For example there may be a simulated low-expression gene that was "differentially expressed" in ground-truth, however, was only expressed in two cells after simulation of the low expressed gene. In this situation, this gene it would not be realistically possible to "detect" this gene as differentially expressed even if ground truth clusters were known. Therefore to generate a background of detectably DEGs, were performed differential expression analysis by 1-way ANOVA (aov function) using the known ground truth cluster labels. This gives us a list of detectably DEGs to use as the ground truth desired genes for feature selection, while non-detectably differentially expressed were all treated as not desired for selection. This parallels pathway analysis in that, if a gene is not detectably expressed, it should not be included in the custom background.

### Anti-correlation-based feature selection performance across various hyperparameters

To assess performance using different selections of the anti-correlation algorithm's FPR and FDR hyperparameters, we assayed three FPRs: 0.1, 0.01, 0.001 (0.001 default) and 6 FDRs: (0.99, 0.5, 0.25, 0.066, 0.1, 0.01) (0.066 is the default). Benchmarking in this context requires a ground truth, leading us to use the Splatter and SERGIO simulations described above, again testing for selection of detectably DEGs between clusters as the desired features.

The metrics used to quantify clustering efficacy of feature selection algorithm were: FPR, TNR, true positive rate, FNR, precision, FDR, false omission rate, accuracy (ACC), balanced accuracy (BA), F1-score, and MCC. These were implemented within the bin/fdr_sweep.py script within the benchmarking repository.

This hyperparameter sweep was also applied to the Splatter simulated null datasets (n = 20; 1 cluster). We quantified the number of selected genes, as well as classifying whether the algorithm "passed" the null-test by returning zero genes; these script for this benchmark is located in bin/param_on_null.py file with, results shown in Supplementary Fig. 3b, c.

### Pancreatic datasets for feature selection

The seven pancreatic datasets[3,26–30] used for feature selection efficacy benchmarking were processed as previously described[3]; the available post-processing datasets were used as-is. These datasets are also now re-packaged in the data zip contained within the benchmark repository. To assess efficacy, three primary metrics were used via gProfiler analysis using the human protein atlas "HPA" pathways which indicates genes are enriched for certain tissues and sub-tissue niches[31,32]. For each dataset, a custom background was used, comprised of the genes expressed in the given dataset. For each analysis, the HPA results were filtered to include only the pancreatic tissues and niches, the pancreatic HPA pathway that was the most significant was counted as a method's best pancreatic match. The -log10($p$-values), precision, and recall for this best match was used for comparisons. To adjust for the wide range and skewed distributions in significance across datasets and methods, we rank transformed the -log10($p$-values); precision and recall however are all on a scale between 0 and 1, and were therefore analyzed directly. Significance was determined with the aov and TukeyHSD functions to measure the main effects and post-hocs respectively. The aov function was called with the formula: metric ~ method + dataset.

### Tabula Muris dataset

The senescent Tabula Muris dataset[33] was used to demonstrate the scalability of our analytic pipeline. This dataset was previously filtered to contain only cells with ≥2500 UMI counts. We therefore

downsampled the dataset such that all cells contained 2500 UMI, and log2 transformed it for analysis. The downsampling process was performed using the bio-pyminer-norm package that is pip installable:

python3 -m pip install bio-pyminer-norm

The process of downsampling is reported in detail at the repository website: https://bitbucket.org/scottyler892/pyminer_norm

Subclustering rounds were first feature selected with the anti-correlation package that we released here, using default parameters:

from anticor_features.anticor_features import

anti_cor_table = get_anti_cor_genes(exprs, feature_ids, species = "mmusculus")

Locally weighted Louvain modularity was used for clustering as described above. Note that while the default functionality of our feature selection package automatically removes ribosomal, mitochondrial, and hemoglobin related genes, for fair comparison with other methods, these genes were left in for possible selection when comparing to other algorithms. This can be customized using the pre_remove_pathways argument. The default removal list are genes contained in the following pathways (all related to ribosomal, mitochondrial, and hemoglobin):

"GO:0044429","GO:0006390","GO:0005739","GO:0005743","GO:0070125","GO:0070126","GO:0005759","GO:0032543","GO:0044455","GO:0005761","GO:0005840","GO:0003735","GO:0022626","GO:0044391","GO:0006614","GO:0006613","GO:0045047","GO:0000184","GO:0043043","GO:0006413","GO:0022613","GO:0043604","GO:0015934","GO:0006415","GO:0015935",      "GO:0072599","GO:0071826","GO:0042254","GO:0042273","GO:0042274","GO:0006364","GO:0022618","GO:0005730","GO:0005791","GO:0098554","GO:0019843","GO:0030492"

Alternatively, if the user whishes to exclude specific features, these can be included in the pre_remove_features list argument; however, this was left empty for all of the work presented here.

## Tabula Sapiens dataset demonstrating Simpson's paradox

The muscle samples from the Tabula Sapiens were subset for only donor TSP2 as this sample was analyzed by 10X Genomics' Chromium and Smart-seq2, thus providing *technical replicates*, therefore removing the possibility of biologically meaningful differences between samples, leaving only the effect of technological differences as a confound to demonstrate that this effect interferes with detection of correlation structures. We next applied our anti-correlation-based feature selection approach (with default parameters), to the subset of cells from each technology alone, quantifying all cross-feature correlations within a technology. Then we performed the same procedure in the combined dataset. We selected candidate Simpson's paradox gene-pairs as those with negative Spearman rho correlations in both datasets when assayed separately, but with a positive correlation when assayed jointly. As an example, we show the scatterplots of *TPT1* and *HSPA1A* (Supplementary Fig. 4).

## 1-million PBMCs

A dataset of 1-million PBMCs was downloaded in mtx format from Parse Biosciences as suggested by the company's website[60]:

https://support.parsebiosciences.com/hc/en-us/articles/7704577188500-How-to-analyze-a-1-million-cell-data-set-using-Scanpy-and-Harmony

We then performed QC measuring the total counts, total mitochondrial counts, total counts mapping to *MALAT1* and *NEAT1* (both nuclear long non-coding RNAs). Cells retained were those which had ln(total-counts)>6.8 and <9.25, percent mitochondrial counts >0.05 and <8.0, and percent nuclear lncRNAs >1.75 and <8.0. These cutoffs were selected interactively with plotting, such that any outlier populations were removed, keeping only cells in the central distribution of these parameters, in a manner similar to selecting populations in flow

cytometry. This process was performed using the bin/pbmcs/do_qc.py, bin/pbmcs/plot_qc.py, and bin/pbmcs/do_filter.py scripts in this repository. Included cells were normalized using the previously described relative log expression method[61], making use of TMM scaling factors[62], as implemented in the bin/pbmcs/do_rle.py script in this repository.

Anti-correlation based feature selection was performed using default parameters, similar to the above described *Tabula Muris* dataset analysis. Genes were then scaled and clipped using the scanpy functions sc.pp.scale(adata, max=10) (default suggestion by scanpy authors). The first 50 principal components (PCs) were taken using the scanpy sc.tl.pca(adata, svd_solver = "arpack", n_comps=50). This PC matrix was then saved as an hdf5; the Euclidean distance on these PCs were taken using the bio-pyminer package[3] with the following command line call:

python3 -m pyminer.mat_to_adj_list -euclidean_dist -hdf5_out -transpose -skip_spearman -hdf5 -i clustering/PCA.hdf5 -ids clustering/pc_names.txt -col_ids clustering/columns.txt -block_size 40000

Clustering was then performed using locally weighted Louvain modularity as described above within the bin/pbmcs/do_louvain.py script, using the pyminer_clust_funcs.do_louvain_primary_clustering function, feeding in the Euclidean distance matrix above.

Coarse grain cluster name identification was facilitated by comparison of average transcriptome signatures with the average transcriptome signatures of a reference PBMC dataset[35]. This was performed by performing Spearman correlations on the subsets of the mean transcriptomes, only including the features that were selected in our original analysis of the 1-million PBMC datasets. The displayed heatmap shows a linear-normalized (between zero and one) for each cluster in question from the 1-million PBMC dataset, thus highlighting the populations for which the average signature was most correlated to within the reference. This analysis is contained in the bin/pbmcs/get_refs.py file of the benchmark repository. Additional labeling of clusters was performed manually based on known marker genes of the noted populations. The code for these exploratory analyses are located in the bin/pbmcs/additional_sub_clust.py, bin/pbmcs/pbmc_abundance.py, and bin/pbmcs/more_plots.py, files within the benchmark repository.

In several cases however, particularly with the classical monocytes, there were more than one cluster from the 1-million PBMC dataset that mapped to a single reference, thus requiring greater investigation of marker genes to identify the biological signal differentiating these clusters. Manual inspection of Wilcoxon differential expression between several clusters showed that in some cases, clusters were split primarily based on sequencing depth; this was determined by one cluster not showing any (or few such as MALAT1) significant up-regulated genes compared to the other. These clusters were then manually merged into single populations: (5,8), (6,9), (12,21), and (13,23,25). These analyses are contained in the bin/pbmcs/more_plots.py script. Wilcoxon DEG analysis was performed in scanpy via the sc.tl.rank_genes function[25].

## Calculating subject pairwise PMD, and residuals

PMD was calculated using the "pmd" function in the pypercent_max_diff python package[47], with source-code available below:

https://bitbucket.org/scottyler892/pypercent_max_diff/src/master/

The PMD calculation call was as follows:

pmd_res = pmd(cluster_by_subject_contingency_table)

in which the cluster_by_subject_contingency_table contains the clusters in rows and subjects in columns.

The pairwise PMD results comparing all subjects (Fig. 3d) to each other is contained in the pmd_res.post_hoc data frame.

The standardized residuals (Fig. 3f) was contained in the pmd_res.z_scores data frame. These values were used for differential

abundance testing as described in the Statistics and Reproducibility section.

## Spring embedding displays

Given that many low dimensional projections can introduce inaccuracies when depicting cell-cell distances[63], we chose to utilize spring embeddings are only intended to represent the undlerying kNN used for clustering, rather than the global or local distances between cells. A 10% subset of million cells were selected for display for computational efficiency. Within each cluster, the top 10% of cells were used as ranked by highest page-rank[64], as these are the most central to the cluster. Page rank was calculated using the networkx "pagerank" function[55]. Spring embedding plots were generated by the pyminer.pyminer_cell_graph_plotting.get_pos call as follows:

$$G = get\_pos(G\_sub, pos\_iters = 1)$$

## Simulations for clustering accuracy

Simulations were performed with both SERGIO and Splatter with the full sweep of clusters and iterations as described above. To systematically vary the degree of signal to noise, we either included all true DEGs (100% signal), or randomly excluded 50% or 80% of true DEGs, leaving a total of 20%, 50%, or 100% of genes contributing to signal, while in all cases, non-DEGs (noise) were all included. To assess cluster performance, we compared cluster results with the ground truth cluster annotations from the simulation. Benchmarking was performed using adjusted Rand index and normalized mutual information as implemented in scikit-learn package[65], while purity and reverse purity functions are contained in the bin/pbmcs/pbmc_abudance.py script in the main repository for this work.

## Statistics and reproducibility

**T1D over/under abundance.** A general linear model was used to test significance on the PMD standardized residuals, comparing T1D to control groups for differing residual z-scores between disease groups, using total counts as a covariate using the statsmodels package with the statsmodels.formula.api function, with R-style formatting:

$$smf.OLS(function = ''clust\_abudnace\_Z \sim disease\_status + total\_counts'')$$

All p-values were all FDR corrected using the Benjamini-Hochberg correction, from the statsmodels package[66] with the statsmodels.stats.multitest.fdrcorrection function.

**Simulations for accuracy.** For displaying metric values, we regressed out the effects of non-displayed variables using the statsmodels ols function as described above, using the fit.resid_pearson residuals for display. For calculating final statistics however, a joint model of all variables was used as shown below:

$$<cluster\_metric> \sim num\_clust + sim + percent\_signal$$
$$+ cluster\_method + fs\_method$$

Resultant *p*-values were all Benjamini-Hochberg corrected for multiple comparisons using the statsmodels statsmodels.stats.multitest.fdrcorrection FDR funcion[66].

## Reporting summary

Further information on research design is available in the Nature Portfolio Reporting Summary linked to this article.

## Data availability

The T1D and health control scRNAseq re-analysis dataset, and the scripts used to generate them are available at figshare https://doi.org/ 10.6084/m9.figshare.22651825. The Tabula Muris Senis dataset is available under the accession GSM4505404. The original dataset is available from the original publication[60] and the Parse Bioscience website [https://support.parsebiosciences.com/hc/en-us/articles/7704577188500-How-to-analyze-a-1-million-cell-data-set-using-Scanpy-and-Harmony]. All other datasets are distributed within the data.zip file within the primary repository: https://bitbucket.org/scottyler892/anti_correlation_vs_overdispersion/. All additional displayed processed data in this study are provided in the Supplementary Information/Source Data file. Source data are provided with this paper.

## Code availability

All code used for implementing the anti-correlation-based feature selection approach is available as a stand-alone package: https://bitbucket.org/scottyler892/anticor_features and is also pip installable: python3 -m pip install anticor_features. All code for running simulations and comparisons used in this study are available at: https://bitbucket.org/scottyler892/anti_correlation_vs_overdispersion/. This benchmarking code is also availble on figshare[67] https://doi.org/10.6084/m9.figshare.23571921.

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

## Acknowledgements
This work was funded by K99 HG011270 (S.R.T.), U01 AG046170 (E.E.S.), and RC2 DK122532 (E.E.S.).

## Author contributions
S.R.T. conceived of and carried out all analyses and wrote the manuscript. E.E.S. guided and supervised work and edited the manuscript. E.G. guided work and edited the manuscript. D.L.O. performed manual analysis of differentially expressed genes in PBMC dataset, identifying the identities of the differentially abundant subsets.

## Competing interests
The authors declare no competing interests.
