## [Peer Review File · Nature Communications]

Anti-correlated Feature Selection Prevents False Discovery of Subpopulations in scRNAseqReviewer #1 (Remarks to the Author):

In this article, the authors introduce a new feature selection method for single cell analysis. The method is based on the anti-correlation patterns to select the useful genes from the original feature sets. The method aims to improve the performance of cell-populations sub-clustering algorithms, as well as to address the issue that the cell-populations cannot be further segmented (i.e., single cell-type/subtype dataset). Overall, the manuscript is well written, and the method is well explained. My concern is that the article is not novel from the computational perspectives. In addition, the validation is weak as authors only test their method on synthetic data. Also, they only compared their method against three methods while there are tens to hundreds of methods that have been developed for single-cell clustering and feature selection. Below are my comments that I hope they can help to improve the quality of the article:

- 1) For the real data analysis, the authors can use the data from single cell database for human, namely Tabula Sapiens (<https://tabula-sapiens-portal.ds.czbiohub.org/>), to test their method. The database is publicly available and contains single-cell data from many validated cell types and their subtypes. At the moment, the authors only test their methods on approximately 250k cells, which has low number of cells compared to today's standard. The authors should extend their analyses to at least 10-20 other datasets with millions of cells.
- 2) The authors only compared their method against 3 other methods, which are too few. I suggest the author compare their method against at least the 20 methods listed in the Genome Biology paper (PMID: 34847932). They should also add the comparison between using their method and using only validated marker genes for the cell types/subtypes.
- 3) The proposed method should work well for general clustering purpose. Other than 3 kNN-based methods, the authors should report the clustering performance of state-of-the-art methods, such as SC3, SEURAT, scCAN, SINCERA, CIDR, SCANPY, and scDHA, with and without using their method for feature selection.
- 4) The authors should provide a figure that describes the workflow of the method. At the moment, the article lacks an illustrative presentation that helps readers to quickly grasp the overview of the method.

Reviewer #2 (Remarks to the Author):

This paper proposed an anti-correlation method to select features for clustering, the idea of anti-correlation is interesting, and the proposed statistical testing method is sound, and the results are appealing. However, I have several comments,

1. To my understanding, the testing procedure for a single cell type and multiple cell types are deferent. However, we donot kown what's the case given a specific data, how to test in the uniform way.
2. To my understanding, the features selected for each round of partition may not be limited to a single gene, how does the number of genes selected affect the clustering result?
3. Clusters can be hierarchical as shown in (Fig. 1 a), and the final result of clustering depending on to what level one may look at the data. Does the anti-correlated features really help to determine the level of clustering. i.e, how do you know the number of cell types should be 3 instead of 2 (Fig 1, a)
4. Given the same set of features, the number of clusters as well as the result of clustering depend on the method used for clustering. How to choose the right method with the help of anti-correlation features.

Reviewer #3 (Remarks to the Author):

The manuscript focuses on clustering of single-cell RNAseq data where the goal is to avoid spurious recursive division of existing clusters. To this end, the authors proposed to only use genes that pass an anti-correlation screening step in any paired clustering algorithm. This approach is motivated by the observation that if a cluster contains two sub-clusters, then the marker genes of the two-clusters should be negatively correlated beyond randomness. The authors proposed a feature selection pipeline based on this observation. To validate the proposed method, the authors showed that (1) on cell line datasets, the features selected by the proposed pipeline prevent further sub-cluster division; (2) on synthetic and real pancreatic data sets, features selected have better interpretations than competing feature selection methods; (3) on a Tabula Muris data, genes selected by the pipeline led to correct identification of a rare population of pancreatic-polypeptide expressing PP-cells.

My major concerns are the following:

(1) Global vs. local anti-correlation screening: The motivation for anti-correlation screening is local to a cluster of cells which could potentially be further divided, while the feature selection pipeline is global which involves all cells from all possible clusters. It is well known that correlation between a pair of variables can be negative across stratum while being positive marginally (a.k.a., Simpson's Paradox). The reverse (negative marginally and positive within each stratum) is also possible for exactly the same reason. Thus, it is not statistically sound to couple the local motivation with the global pipeline. Of course, there will be genes that are negatively correlated both marginally and within each stratum. However, the current pipeline, as proposed, suffers from both more false positives and more false negatives for the reason stated above. Ideally, the procedure should be applied every time a potential further sub-cluster division is considered, but this comes with the cost of more computation and fewer cells as the recursive division gets deeper. A compromise among false discovery rate, detection power and computation cost needs to be made.

(2) Sensitivity to tuning parameter choices and batch effects: Take the current pipeline as is, the authors did not test carefully how sensitive it is to batch effects commonly seen in scRNA-seq data and to choice of tuning parameters. A more systematic evaluation is necessary to justify the practicality of the proposed method.

(3) Power in detecting fine clusters: The validation data sets the authors considered all have clearly defined clusters, such as those related to pancreas. Can this pipeline be used in distinguishing more subtle cell types, such as various subtypes of CD8 T cells, and naive B, intermediate B, and memory B cells? Understanding these finer scale cell populations is of concurrent interest, and it would be great if the authors could demonstrate how the proposed pipeline works on these cases.

Reviewer #1 (Remarks to the Author):

In this article, the authors introduce a new feature selection method for single cell analysis. The method is based on the anti-correlation patterns to select the useful genes from the original feature sets. The method aims to improve the performance of cell-populations sub-clustering algorithms, as well as to address the issue that the cell-populations cannot be further segmented (i.e., single cell-type/subtype dataset). Overall, the manuscript is well written, and the method is well explained. My concern is that the article is not novel from the computational perspectives. In addition, the validation is weak as authors only test their method on synthetic data. Also, they only compared their method against three methods while there are tens to hundreds of methods that have been developed for single-cell clustering and feature selection. Below are my comments that I hope they can help to improve the quality of the article:

General comments to Reviewer 1: Many of the comments brought up by reviewer 1 point to truly important challenges in our field; we thank the reviewer for highlighting these issues. We believe that in addressing the issues raised by the reviewer has both improved the clarity and scope of our work.

Rev1-Q1: For the real data analysis, the authors can use the data from single cell database for human, namely Tabula Sapiens (<https://tabula-sapiens-portal.ds.czbiohub.org/>), to test their method. The database is publicly available and contains single-cell data from many validated cell types and their subtypes. At the moment, the authors only test their methods on approximately 250k cells, which has low number of cells compared to today's standard. The authors should extend their analyses to at least 10-20 other datasets with millions of cells.

Rev1-R1.1: Overall, we agree with the reviewer, that a demonstration using larger datasets may be beneficial to show the scalability of our approach. In total, we have now included an analysis of 15 real-world datasets, which include millions of cells. These datasets include 2 cell lines, 7 human pancreatic datasets, 4 datasets used for recursion to completion (mouse and human, single cell and nucleus), the 245k mouse atlas dataset, and 1-million PBMCs from 24 donors. The analysis of this last dataset led to our discovery of increased activated classical monocytes, naïve CD4+ T-cells, effector-memory (EM) CD4+ T-cells, and cytotoxic CD8+ T-cells in Type-1 Diabetes. This sums to a total of 15 real-world datasets with millions of cells analyzed.

Figure 3: Application to large real-world datasets. **a**, A heatmap of the top 5 marker genes per cluster are shown for the primary lineages from the full senescent *Tabula Muris* dataset³³, with the last cluster representing a mixture of cell-types from the endocrine pancreas. **b**, When subclustered with anti-correlated feature selection, cell-type droplets (x) as well as classically described leukocyte, α , δ , β , and acinar populations were discovered. Subclustering β cells discovered mixed-lineage droplets with δ and leukocyte cells as well as the rare PP-cell population, but additional subclustering of PP-cells was prevented by anti-correlation-based feature selection. **c**, Selected features for clustering 1-million PBMCs. **d**, Subject-level reverse Percent Maximum Difference (rPMD), shows that Type-1-Diabetes (T1D) subjects are more similar to each other, while control PBMCs are more diverse by cluster composition. **e**, A spring

embedding of a subset of cells from each cluster, color-coded by donor, with sub-plots for T1D and control subjects, showing large-scale uniformity in T1D compared to the heterogeneous control samples. Note that this is for display purposes only, was not used in analysis, and does not represent cell-cell distances, but rather a display of the graph used for clustering. **f**, A heatmap of PMD standardized residuals, which correspond to the significance of how different each subject's relative abundance of all clusters differs from the null expectation of no-difference between subjects. A matching bar-chart shows the T-statistic of cluster level significance for each cluster's differential over-under abundance shown in the heatmap, comparing T1D to controls. Bars are color-coded by significance ($P < 0.05$ after Benjamini-Hochberg). **g**, The spring embedding of the kNN graph is color-coded by significance of differential abundance for each cluster, and additionally color-coded by T1D/control status, then again subset for only the significant clusters.

We also want to provide more context as to why we selected this approach, instead of using the labels provided in the publication of the *Tabula Sapiens* dataset (which contains 500k cells). There are three points of concern discussed below (in **Rev1-R1.2**, **Rev1-R1.3**, and **Rev1-R1.4**) regarding measuring consistency with the published labels as a measure of accuracy. The first is Simpson's paradox that can result by jointly analyzing datasets from differing technologies (as in the *Tabula Sapiens*) (**Rev1-R1.2**). The second is the human error contained within these published labels (**Rev1-R1.3**). The final reason is that by benchmarking for consistency with published labels, one would be benchmarking for similarity to the processing steps used by the original authors (scVI and UMAP dimension reduction in the case of the *Tabula Sapiens*) (**Rev1-R1.4**). This last issue could cause us to end up in a self-reinforcing loop by creating benchmarks that incentivize consistency with standard methods rather than benchmarking for accuracy (which we admit is a very challenging question, that still requires much attention within the field).

Rev1-R1.2: The *Tabula Sapiens* dataset was collected from different individuals often using different technologies. This is a strength in some respects, but a limitation in others. Given that each technology will have its own strengths and weaknesses, it is in our view an overall strength that the *Tabula Sapiens* consortium was built using two different technologies. However, the introduction of the compounded effects of differing technologies can introduce artifactual correlations that will impact the ability to use correlation structure for feature selection.

If differing technologies are analyzed jointly, this has the potential to give rise to correlations that are a consequence of the mixed technical and biological effects (the correlation exists in the jointly analyzed set but not in either of the individual sets, a situation referred to as Simpson's Paradox). This issue was discussed by Reviewer 3. The introduction of *technical correlations* that are artifacts based on a joint analysis across different technologies, is problematic however. To address this, we have now added a figure discussing the potential implications of confounding technical and biological effects, in the context of applying our newly proposed method.

Extended Data Figure 4: Simpson's Paradox may be introduced with multi-technology analyses. An example scatter plot of two genes (*TPT1* and *HSPA1A*) from technical replicates from the *Tabula Sapiens* dataset processed in parallel using Smart-seq2 and 10X Genomics' Chromium chemistry, shows global negative correlations when the two patterns are assessed independently with respect to each technology. However, when combined, a much more significant positive correlation emerges, a clear artifact from confounding the technical and biological effects, with two negatively correlated biological effects converted to a strong positive correlation given the domination of the technical effects when the data across technologies are combined.

The next potential issues that could arise in using previously published cell-type and subtype labels is simple human error:

Rev1-1.3: The second issue in re-using previously published labels as ground truth is perhaps simpler: human error. We have done in depth analyses of the *Tabula Sapiens* dataset, and highlight an instance showing obvious errors, in which the experimenter's expectation in terms of which cell types should be present in a given tissue, has biased the labeling. In the example below, we show that B-cells were mis-identified as skeletal muscle satellite stem cells, yet they were topologically merged within a largely uterine fibroblast cluster.

Example error in the *Tabula Sapiens* atlas due to tissue specific priors. **a**, UMAP coordinate showing the *tabula sapiens* stromal compartment, color-coded by donor. We focus on donor TSP2. **b**, UMAP color-coded by the custom annotations provided by tissue experts; the highlighted subset is the region of interest with the donor of interest highlighted; these droplets were labeled as muscle satellite stem cells, while other cells within this topology were largely labeled as endometrial or uterine fibroblasts. **c**, The tissue of origin shows the droplets of interest were from a muscle sample, likely explaining their label of muscle satellite cells. Color-coded expression for B-cell marker genes shows that these droplets are consistent with B-cells,

yet are topologically integrated with endometrial/uterine fibroblasts. **d**, The droplets of interest were also negative for the fibroblast marker *Col1a1*. **e**, Droplets of interest were also negative for the satellite stem cell marker *Pax7*, despite many of the cells within the larger satellite cell cluster being positive for this marker.

Similar errors in the MNIST dataset have also been described (<https://doi.org/10.48550/arXiv.1911.00068>); however with single cell RNAseq datasets, we cannot simply look at the original data and instantly realize that a “1” was labeled a “9.” This unfortunately leaves us comparatively blind to where and when these errors occur in the mega-atlases built on merged datasets across differing technologies.

The third issue that can arise in using previously published labels is that we incentivize similarity to the methods used for processing the dataset.

Rev1-R1.3: By benchmarking for similarity between published class labels and unsupervised clustering labels, we in essence accept as “ground truth” the approach used by the original authors. In the case of the *Tabula Sapiens* atlas, the methods used were integration with scVI on raw counts (without the use of any feature selection algorithm), and a latent dimension mapping via UMAP using the default settings (again with no feature selection). In essence, one could achieve a perfect score by copying and pasting the code from their python notebook (https://github.com/czbiohub/tabula-sapiens/blob/master/Fig1/Train_scVI.ipynb). Yet, making an arbitrary change to the hyperparameters in the UMAP projection, or use of a feature selection algorithm, would have resulted in differing apparent topologies for the tissue annotators to label; however “ground-truth” labels should not depend on the selection of hyperparameters or the selection of preferred algorithms and processing steps by the prior authors, who published the original cluster labels.

Building on this, in Greg Gibson’s perspective piece (Director: Center for Integrative Genomics, Georgia Tech), he wrote: “In my group’s experience, it is not unusual for reanalysis to find 20% fewer or more clusters in datasets downloaded from public repositories, with between 50% and 70% equivalence of cell-type assignments.” (PMID: 35536863). With this lack of reproducibility in mind, we are cautious in utilizing previously published labels as “ground-truth.”

In summary, there exist technical issues with using multi-technology, non-multiplexed, atlases for benchmarking in this context, we have found errors in these labels, and benchmarking for consistency may not correlate with accuracy. Analysis of multi-technology atlases (as with the *Tabula Sapiens*, which was analyzed without utilizing any feature selection) can introduce Simpson’s paradox due to the hierarchical effects of technical and biological signal, which would give erroneous signal in the context of our feature selection algorithm. We have shown above in **Rev1-R1.3**, that there are also notable errors, in part traceable to both inaccuracies in integrations paired with the use of prior expectation by manual annotators. Lastly, the use of the labels generated by one group, incentivizes similarity to the approaches used by that group, when that approach is established as ground truth, despite the wide variability observed across different analysts with respect to assigning labels to clusters. For these reasons, we prefer to avoid using consistency with the labels generated by others for the purposes of benchmarking, despite this being an approach frequently used in the literature.

Rev1-Q2: The authors only compared their method against 3 other methods, which are too few. I suggest the author compare their method against at least the 20 methods listed in the Genome Biology paper (PMID: 34847932). They should also add the comparison between using their method and using only validated marker genes for the cell types/subtypes.

Rev1-R2: We have now added additional methods (7 in total), however, more importantly, we perhaps did not clearly enough articulate the class of methods relevant for comparison to our method, which involve the distinctions between feature selection in an unsupervised vs. supervised setting as described more fully below. We have thus added text to our manuscript describing the

Schematic Figure Showing Primary and Secondary Analysis Pipelines:

The top panel (grey) shows a primary analysis pipeline, for which one of the earliest steps is unsupervised or supervised feature selection to identify either unknown groups based on intrinsic structure of the data, or supervised “known markers” to identify the expected groups. Here we benchmark only the methods for unsupervised feature selection in primary analyses (yellow outline), note that our new method (pink outline) is conceptually different than other previous methods that test for how well single-gene distributions match expectation vs gene-gene structure with our method. Notably, many more methods exist for secondary analyses (bottom orange panel) including analyses for feature selection (green) upstream of a label transfer classifier. These are algorithmically distinct, and downstream of feature selection in a primary analysis. Both classes of “feature selection” are highlighted in green panels.

subtle differences between these approaches and better justifying the methods we selected for comparison (quoted at the end of this section), and have also added to the algorithms tested here.

The term “feature selection” has unfortunately been re-used for slightly different purposes. The term “feature selection” has been applied to selecting the most meaningful features to use in a classifier. However, “feature selection” can also indicate an upstream processing step prior to unsupervised clustering; this later purpose is the form of feature selection used in this work. Ultimately, we think that the field may benefit from a review article to clarify explicitly the ways in which some of our field’s terms are re-used.

In all cases, feature selection identifies a subset of all input features that are relevant to a problem; however, there are two classes of feature selection algorithms (noted by green panels in the above figure). For feature selection in supervised classifiers (bottom panel), the features are selected based on their ability to discriminate samples for the target classifier label; this is the form of feature selection discussed in the review noted by the reviewer (PMID: 34847932). This means that the features are selected based on their statistical relationship with the target variable (user-defined cluster labels in this case). The goal of feature selection in this context is to improve the performance of the cell-type classifier label transfer by reducing the dimensionality of the feature space, while preserving the discriminatory power of the features. Indeed, the single-cell packages discussed in the review (scClassify, ACTINN, CHETAH, CaSTLe, Garnett, Moana, scID, scPred, scmap, singleCellNet, LinearSVC) are for label transfer, in which a “reference dataset” is used to train an ML classifier algorithm, with a feature selection algorithm used to increase accuracy and speed of the label transfer model.

However, when the downstream application is unsupervised clustering (top panel), the goal of feature-selection is identifying features that best capture the intrinsic structure of the data, removing less structured features that contribute noise, allowing the unsupervised clustering algorithm to identify meaningful groups. In this way, unsupervised feature selection prior to unsupervised clustering differs from feature selection approaches in supervised label transfer as discussed above. For label transfer with classifiers (bottom), these assess for a relationship between gene expression and the target labels in the reference dataset, and can primarily re-discover the labels present in the reference data.

With unsupervised feature selection upstream of unsupervised clustering, the target variables (clusters) are *not known*, and the objective is to discover novel groups and identify the underlying structure of the data. Both techniques are tremendously useful, but serve different purposes, and therefore the feature selection algorithms for label transfer described in the noted review (PMID: 34847932) are not really relevant to the domain of unsupervised feature selection for the purpose of unsupervised clustering in a primary analysis. In fact, the feature selection algorithms used in a secondary analysis such as with label transfer, build on, and require, the cluster labels generated through methods contained in the “primary analysis” panel of the above figure, which includes the use of unsupervised feature selection algorithms that we are indeed benchmarking (yellow border, top primary analysis panel).

In comparison to feature selection for classifier performance, unsupervised feature selection for unsupervised clustering is comparatively under-researched. Our original comparison utilized 5 different approaches to unsupervised feature selection (1 being our new method, 3 related to the mean-variance relationship, and 1 related to the mean-dropout relationship). However, we have now added deviance (from “expected” distribution) and zero-inflation, to now cover 7 different methods of feature selection for unsupervised clustering.

We have also added to the text to clarify the distinction between the two different classes of feature selection, noting that we were investigating only those upstream of unsupervised analyses:

Lines 26-32

A frequent first task in performing cell-type identification from scRNAseq is feature selection to identify genes that are cell-type specific markers based on various statistical properties, prior to unsupervised clustering. These algorithms differ from feature-selection applied in the context of a classifier for cell-type label transfer¹. Current approaches to feature selection prior to unsupervised clustering in single cell -omics include measures of the relationship between a gene's mean and variance (i.e., overdispersion)²⁻⁴, a gene's mean and dropout rate⁵, a gene's deviance from an expected distribution^{6,7}, or degree of zero-inflation⁷.

Rev1-Q3: The proposed method should work well for general clustering purpose. Other than 3 kNN-based methods, the authors should report the clustering performance of state-of-the-art methods, such as SC3, SEURAT, scCAN, SINCERA, CIDR, SCANPY, and scDHA, with and without using their method for feature selection.

Rev1-R3: We thank the reviewer for this suggestion; we have provided an additional benchmark to examine the performance of the various clustering algorithms both with and without our anti-correlation-based feature selection:

Clustering algorithms: our original implementation of locally weighted Louvain modularity, Sc3s, Seurat's Leiden implementation, scCAN, SINCERA, Scanpy's implementation of Louvain clustering, Scanpy's implementation of Leiden modularity, and scDHA. While we tried to use CIDR as well, there was an irreconcilable broken dependency (minpack.lm), even after reaching out to the minpack.lm developer.

Feature selection parameters: As requested, we provided a comparison with no feature selection, feature selection with our pipeline under default parameters (FDR=0.001), and with our pipeline under a more sensitive FDR=0.25 setting, where we found empirically that the range (0.001-0.25) yielded valid results as well.

Differing signal to noise ratios: We also sought to characterize performance with different levels of signal to noise. To this end, we masked differing percentages of ground truth DEGs upstream of feature-selection, while including all noise genes. This changes the relative ratio of signal contributing to noise-contributing-genes. We used three settings (20%, 50%, or 100% of original signal).

Simulation parameters: Both SERGIO and Splatter simulations were used, simulating 4, 6, 8, and 10 clusters permuted with all of the above described variable classes.

As we discussed in **Rev1-R1**, for the purpose of identifying correct ground-truth clusters, we performed this assessment on simulations, in which the ground truth can be directly controlled, rather than relying on similarity to previous pipelines for more complex datasets. While most methods appeared to perform well, the top three performers were: locally weighted Louvain, Seurat's Louvain, and scCAN. We also found that use of our feature selection algorithm did indeed increase cluster accuracy as measured by adjusted Rand index, normalized Mutual information, purity, and reverse purity. The results of this comparison are shown below:

Extended Data Figure 4: Clustering algorithm performance with and without anticorrelation-based feature selection. Scatter plots showing the relationship between signal to noise in the input (x-axes) and how clustering algorithm and feature selection algorithms impact clustering performance as measured by four metrics (y-axes). In all cases, y-axes show the residuals of a linear model fit after regressing out all other variables, not of interest including number of clusters simulated, simulation program. Y-axes indicate the values of metrics noted in the panel titles, after correction for co-variables. Using Splatter and SERGIO simulations, we benchmarked cluster algorithm performance with no feature selection, feature selection with the default FDR=0.001 hyperparameter, and feature selection with a more sensitive hyperparameter setting (FDR=0.25). Simulations were performed with 3 different values of signal to noise: 1) all (100%) true differentially expressed genes (DEGs) between clusters were included in the input, 2) 50% included, and 3) 20% included, while retaining all randomly expressed genes in all cases, thus varying the input dataset's signal-to-noise ratio. Displayed points are either all individual measures across all simulations (small spots to the right), or the

median (large dots to the left). All non-displayed factors were regressed out for display purposes, but statistics were calculated on a joint model using a mixed-effects linear model.

Rev1-Q4: The authors should provide a figure that describes the workflow of the method. At the moment, the article lacks an illustrative presentation that helps readers to quickly grasp the overview of the method.

Rev1-R4: We agree this would readers more quickly digest the paper. In fact, along with a comment from Reviewer 3, we realized that there was a need for a better schematic figure of our recursive clustering analysis approach. To this end, we have now restructured **Fig. 2a** to show a pipeline schematic that better communicates how our feature selection method is used for halting the recursive clustering pipeline when no additional sub-clusters exist.

Figure 2: Recursion-to-completion in real datasets and anti-correlation algorithm scaling. **a**, A schematic of sub-clustering is shown in the form of UMAP projections of the original dataset (left panel), and a sub-clustering iteration of a population found in the first round of feature selection and clustering (right panel). **b-c**, In real datasets of varying technologies, status quo algorithms fail the recursion-to-completion problem while the anti-correlation-based approach prevented recursion-to-completion. Recursive clustering plots where each point

indicates a cluster at a given recursive clustering recursion-depth as denoted in successive rings and color. **d**, Boxplots of the mean recursion depth for each of the final sub-clusters for each noted method. **e**, Boxplots of the total number of groups obtained through iterative sub-clustering.

Reviewer #2 (Remarks to the Author):

This paper proposed an anti-correlation method to select features for clustering, the idea of anti-correlation is interesting, and the proposed statistical testing method is sound, and the results are appealing. However, I have several comments.

General comments to Reviewer 2: We greatly thank the reviewer for their time and effort in reviewing our manuscript. We believe that the added figures and modified text that we have made in response to these comments have greatly improved our work.

Rev2-Q1: To my understanding, the testing procedure for a single cell type and multiple cell types are deferent. However, we donot kown what's the case given a specific data, how to test in the uniform way.

Rev2-R1: We apologize for the lack of clarity on this point. We have clarified that our procedure is actually a single process, whether performing feature selection of the full dataset, or executing the recursive sub-clustering approach. In both cases, if zero features are selected, this is taken to mean that no clusters (or sub-clusters) are present. When performing a single round of clustering, or performing sub-clustering, the procedure is the same and terminates when either the whole dataset is determined to contain only a single cluster, or when a sub-population should not be divided anymore.

Lines 116-121

“This repeated process of feature selection holds three benefits over maintaining the original features with altered cluster resolution: 1) It allows us to use an ‘empty’ feature list as an indication that no more clusters exist (the same method we used for passing the null-test), 2) allows for dynamic detection and subsequent refinement of compound correlation structures that differ at the global and local scale, and 3) does not incorporate noise from features enriched in unrelated lineages.”

Additionally, we hope that the added recursive clustering schematic figure (new **Fig. 2a**) helps further clarify our process.

Figure 2: Recursion-to-completion in real datasets and anti-correlation algorithm scaling. **a**, A schematic of sub-clustering is shown in the form of UMAP projections of the original dataset (left panel), and a sub-clustering iteration of a population found in the first round of feature selection and clustering (right panel). **b-c**, In real datasets of varying technologies, status quo algorithms fail the recursion-to-completion problem while the anti-correlation-based approach prevented recursion-to-completion. Recursive clustering plots where each point indicates a cluster at a given recursive clustering recursion-depth as denoted in successive rings and color. **d**, Boxplots of the mean recursion depth for each of the final sub-clusters for each noted method. **e**, Boxplots of the total number of groups obtained through iterative sub-clustering.

Rev2-Q2: To my understanding, the features selected for each round of partition may not be limited to a single gene, how does the number of genes selected affect the clustering result?

Rev2-R2: Yes, indeed, the reviewer is correct in this interpretation, for two reasons. First, the central principal that differentiates our new method from previous methods is the use of *between gene* comparisons. By examining the structure contained between two genes, our approach is able to pass the null- and recursion-to-completion- tests. But as noted by the

reviewer, this also means that a single gene may distinguish sub-populations if it is expressed randomly in relation to all other genes; however, if there does exist mutual exclusive structure between this gene, and others, then it will be included in the anti-correlation results.

The second reason that a single gene on its own would not contribute sufficient signal to be included is that genes selected in our process must also have significant positive correlations with at least 10 genes; this is built from the premise that a biologically meaningful source of variation should be modulated gene-sets/-modules rather than only a single gene, particularly as single gene measures in single-cell datasets can be exceptionally noisy and provide simply spurious signal.

On the question of how the number of transcripts selected impacts the clustering result, the answer is somewhat complex: Ultimately, the number of features selected by our method depends on the intrinsic complexity of the input dataset. This differs strongly from other methods where the number of features used is frequently a user-defined hyperparameter, which therefore creates a scenario in which the user's hyperparameter is determining the level of complexity detectable by the clustering algorithm. With our method however, the degree of structure in the dataset is what defines the number of features, which allows the downstream clustering algorithm to detect greater levels of complexity with a greater number of features, whereas where less structure exists, fewer transcripts are selected, allowing for detection of the lesser complexity that exists in the input.

For example (with plot below), at a global scale in the context of a full-body multi-tissue atlas, nearly half of the transcriptome was selected by our method as showing meaningful signal. However, as sub-clusters are defined at a more granular level, locally within the dataset, the lineage transcripts will no longer contribute to meaningful signal as they ultimately give way to the few transcripts that provide the meaningful structure for the subcluster. This can be seen below in the decreasing number of selected features (y-axis) in the recursive clustering of the *Tabula Muris* dataset, where the left part of the graph reflects the full body multi-tissue global scale, and the right (third round clustering) reflects the more granular structure (the PP-cell subset in this case) in which we no longer find meaningful structure. This contrasts with standard methodologies that simply use the top 2000 transcripts (by the metric of choice: most commonly the mean-variance relationship, noted in the dashed horizontal line). The recursive clustering level shown below matches those in **Fig 3a-b** in the recursive clustering of the *Tabula Muris* dataset. This highlights the fact that, when using our method, the degree of structure in the input dataset (or local subset of the data) is related to the number of features automatically detected. However, using standard methodologies, the number of clusters or degree of complexity is in fact unrelated to the number of features selected, frequently using the hyperparameter of 2000 transcripts.

Relationship between dataset complexity, and number of selected features: Two line-graphs showing on the x-axis, the recursive clustering round of the *Tabula Muris* multi-tissue dataset, and on the y axis the (a) number of features, or (b) the natural log of the number of features selected by the anti-correlation-based feature selection algorithm. The dashed line shows 2000 transcripts, which has become a widely adopted default number of features to use. Instead of using a static number of features, our implementation enables an automatically determined number of features to be selected, only contributing anti-correlated structure to the current subset of data, becoming progressively more local with greater sub-clustering granularity.

Rev2-Q3: Clusters can be hierarchical as shown in (Fig. 1 a), and the final result of clustering depending on to what level one may look at the data. Does the anti-correlated features really help to determine the level of clustering. i.e, how do you know the number of cell types should be 3 instead of 2 (Fig 1, a).

Rev2-R3: Indeed, this is an important point to make. We now emphasize in the text (see quoted text below) that the selection of anti-correlated features will not determine the level of clustering. We are additionally currently working on answering the question “how many clusters?” for a separate manuscript, which we hope to release soon.

Lines 208-211

“As seen in the final sub-cluster round, however, while anti-correlation-based feature-selection is biologically accurate and answers the question: “Should this cluster be sub-clustered?”, it does not ensure that downstream algorithms will select the correct number of clusters; this remains an outstanding problem as previously reported¹².”

Rev2-Q4: Given the same set of features, the number of clusters as well as the result of clustering depend on the method used for clustering. How to choose the right method with the help of anti-correlation features.

Rev2-R4: This is indeed an important question, and is related to the previous question (Rev2-Q3). We have now added a new benchmark using anti-correlation-based feature selection with 8 different clustering algorithms, that we believe will help the field in identifying best practices for clustering algorithms as well:

Extended Data Figure 4: Clustering algorithm performance with and without anticorrelation-based feature selection. Scatter plots showing the relationship between signal to noise in the input (x-axes) and how clustering algorithm and feature selection algorithms impact clustering performance as measured by four metrics (y-axes). In all cases, y-axes show the residuals of a linear model fit after regressing out all other variables, not of interest including number of clusters simulated, simulation program. Y-axes indicate the values of metrics noted in the panel titles, after correction for co-variates. Using Splatter and SERGIO simulations, we benchmarked cluster algorithm performance with no feature selection, feature selection with the default FDR=0.001 hyperparameter, and feature selection with a more sensitive hyperparameter setting (FDR=0.25). Simulations were performed with 3 different values of signal to noise: 1) all (100%) true differentially expressed genes (DEGs) between clusters were included in the input, 2) 50% included, and 3) 20% included, while retaining all randomly expressed genes in all cases, thus varying the input dataset's signal-to-noise ratio. Displayed points are either all individual measures across all simulations (small spots to the right), or the

median (large dots to the left). All non-displayed factors were regressed out for display purposes, but statistics were calculated on a joint model using a mixed-effects linear model.

Reviewer #3 (Remarks to the Author):

The manuscript focuses on clustering of single-cell RNAseq data where the goal is to avoid spurious recursive division of existing clusters. To this end, the authors proposed to only use genes that pass an anti-correlation screening step in any paired clustering algorithm. This approach is motivated by the observation that if a cluster contains two sub-clusters, then the marker genes of the two-clusters should be negatively correlated beyond randomness. The authors proposed a feature selection pipeline based on this observation. To validate the proposed method, the authors showed that (1) on cell line datasets, the features selected by the proposed pipeline prevent further sub-cluster division; (2) on synthetic and real pancreatic data sets, features selected have better interpretations than competing feature selection methods; (3) on a Tabula Muris data, genes selected by the pipeline led to correct identification of a rare population of pancreatic-polypeptide expressing PP-cells.

General comments to Reviewer 3: We thank the reviewer for their thoughtful comments. They frequently cut directly to the underlying complexities of these datasets. We think that the greater discussions stemming from their comments have indeed greatly strengthened our work, and provide the community with a better context in which to use our new tool.

My major concerns are the following:

Rev3-Q1: Global vs. local anti-correlation screening: The motivation for anti-correlation screening is local to a cluster of cells which could potentially be further divided, while the feature selection pipeline is global which involves all cells from all possible clusters. It is well known that correlation between a pair of variables can be negative across stratum while being positive marginally (a.k.a., Simpson's Paradox). The reverse (negative marginally and positive within each stratum) is also possible for exactly the same reason. Thus, it is not statistically sound to couple the local motivation with the global pipeline. Of course, there will be genes that are negatively correlated both marginally and within each stratum. However, the current pipeline, as proposed, suffers from both more false positives and more false negatives for the reason stated above. Ideally, the procedure should be applied every time a potential further sub-cluster division is considered, but this comes with the cost of more computation and fewer cells as the recursive division gets deeper. A compromise among false discovery rate, detection power and computation cost needs to be made.

Rev3-R1: The reviewer raises a number of important issues, including Simpson's paradox and hierarchical correlation/anti-correlation structures, which we had in fact taken into account when we developed this algorithm. While we had not discussed it explicitly in the original manuscript, given the importance of this point, we now include a brief discussion and demonstration of Simpson's Paradox in scRNAseq data. Further, we note that the process proposed by the reviewer (iterative feature selection within each sub-cluster at the outset of every potential new sub-division), is in fact the procedure we had used in all analyses. Given this was not clearly enough articulated, we have now added a clearer schematic of this process (**Figure 2a**). Our approach indeed begins globally (using the full dataset), and becomes more and more local, upon further subdivisions, therefore ultimately capturing both global and local structures because at each subdivision, the anti-correlation based feature selection process was applied anew, only on the current subset.

Figure 2: Recursion-to-completion in real datasets and anti-correlation algorithm scaling. **a**, A schematic of sub-clustering is shown in the form of UMAP projections of the original dataset (left panel), and a sub-clustering iteration of a population found in the first round of feature selection and clustering (right panel). **b-c**, In real datasets of varying technologies, status quo algorithms fail the recursion-to-completion problem while the anti-correlation-based approach prevented recursion-to-completion. Recursive clustering plots where each point indicates a cluster at a given recursive clustering recursion-depth as denoted in successive rings and color. **d**, Boxplots of the mean recursion depth for each of the final sub-clusters for each noted method. **e**, Boxplots of the total number of groups obtained through iterative sub-clustering.

We discuss and display the potential difficulties with Simpson's Paradox.

Rev3-Q2: Sensitivity to tuning parameter choices and batch effects: Take the current pipeline as is, the authors did not test carefully how sensitive it is to batch effects commonly seen in scRNA-seq data and to choice of tuning parameters. A more systematic evaluation is necessary to justify the practicality of the proposed method.

Rev3-R2: We of course agree with the reviewer that batch effects are a critical issue and in fact is among the most discussed issue in our group when dealing with single cell RNAseq data. Batch effects are well known to add a layer of (unwanted) complexity. Simpson's paradox will again be at play here if all batches are analyzed jointly, as simply changing the distributions between batches, or depth effects can destroy, alter, or induce pseudo-correlations based on technical effects. We have now added a new figure dedicated explicitly to Simpson's paradox and it's role in batch effect mediated difficulties in applying our method (**Extended Data Figure 4**).

Extended Data Figure 4: Simpson's Paradox is introduced with multi-technology analyses. An example scatter plot of two genes (*TPT1* and *HSPA1A*) from technical replicates from the Tabula Sapiens processed in parallel using Smart-seq2 and 10X Genomics' Chromium chemistry shows global negative correlations when the two patterns are assessed within a technology. However, when combined, a much more significant positive correlation emerges, strictly from compounding these technical effects over two negative correlations.

We have also included an explicit discussion of this topic within the main text, including the possibility of biologically derived Simpson's paradox, and the shift from detecting global to local structures with recursive clustering as described in our new schematic (also noted above).

Lines 214-228

One real-world scenario that can be encountered, however, is the use of multiple batches or technologies simultaneously. If appropriate caution is not exercised, such situations can introduce Simpson's paradox, which can result if technical effects are layered on top of the biological effects, changing the overall global correlation structure as a result of mixing two different local correlation structures. Indeed, examining a single donor sample from *Tabula Sapiens* whose cells were assayed in parallel using two differing technologies (10X Genomics' Chromium and SMART-seq2), these technical replicates both showed negative correlations when assayed independently with respect to the two technologies employed. However, when analyzed jointly, these negative correlations became positive, giving rise to Simpson's paradox, despite the fact that these were technical replicates (Extended Data Fig. 5). While common defaults in pipelines will take the intersection of feature-selection runs performed on each individual batch, this will only capture the intersection of biological effects, therefore leaving users at risk of discarding batch confounded, but biologically meaningful variation. This highlights the need to perform feature selection analysis without confounded technical and biological signal.

Another important note is that these effects do not seem to be a major area of concern when using multiplexed data. We show this by our analysis of 1-million PBMCs (12 controls and 12

type-1-diabetes (T1D)), performed in a single batch in multiplex. We unexpectedly observed that T1D PBMCs appeared to be extremely homogenous in their cluster composition, particularly when compared to controls (all co-processed in the same batch), which was both a biologically interesting finding, and confirmation that multiplexing alleviates these issues.

Subset of Figure 3: Spring embeddings of 1-million PBMCs. scRNAseq data represented as a spring-embedding of the kNN used for clustering on type-1-diabetes (T1D) and control samples, color-coded by sample ID. Note that positions are within a unified projection and were simply subset into two different panels to show T1D and controls separately.

Note that no alignment-style batch correction methods were used on any of these data. Rather the data were normalized using with relative log expression (RLE), similar to that included in the Scater single cell package (PMID: 28088763); although we re-implemented these functions in python to work with the scanpy object for this work.

Had the T1D samples been equally heterogeneous as with the control samples, we would be unsure over which attributes of this heterogeneity were technical or biological in nature; however seeing that within a disease homogeneity exists, even across distinct biological samples and donors, to us indicates that the heterogeneity within the controls is likely biologically real, and that technical confounds and concerns over residual technical effects overlayed on biological effects introducing Simpson's paradox are likely not an issue when multiplexing in this manner.

Hyperparameter Sweep

The choice of hyperparameters is always of paramount importance. It was for this reason that we designed our algorithm to always utilize bootstrap shuffled null distributions wherever possible, to prevent the need for a user-defined hyperparameter. There are however two user-exposed hyperparameters: 1) the FPR corresponding to making the call on whether a correlation is significant or not, and 2) the FDR cutoff used for gene selection. The later hyperparameter effectively maps a multiple for the number of significant negatively correlated genes above background relative to the null expectation, as a way to call a gene as having "too many" significant negative correlations, relative to the total number of negative correlations.

Notably, as described in detail in the methods, our FDR cutoff is not a traditional formulation, but rather an FDR that is set given a fixed FPR (0.001). For this reason, the specified FDR and FPR hyperparameters do not actually directly correspond to the feature selection FDR and FPR, for feature selection accuracy.

We now provide a systematic sweep across the FDR parameter assessing performance with Splatter and SERGIO simulated datasets, as these can provide a proper ground-truth and

confusion matrix. We were indeed pleased to see that our default parameters (FPR=0.001 & FDR=.0666: 1/15) performed at or near optimal performance with respect to nearly all metrics, with the potential for only small gains in sensitivity (with Splatter), but were at or near top performance across the board for the SERGIO simulations.

Extended Data Figure 3: Hyperparameter selection with anti-correlation-based feature selection. Scatter plots of 11 different machine learning performance metrics are shown for a hyperparameter sweep of the FDR and FPR settings when running the anti-correlation algorithm. Three FPR settings were used, and are broken into three vertical panels per metric employed (FPR=0.001 (default setting), 0.01, 0.1). The x-axis of all plots show the other parameter (FDR), for which 6 values were used (0.01, 0.1, 0.066 (default), 0.25, 0.5, 0.99). Plots are color-coded by the simulation mode (SERGIO and Splatter). Means of each metric are plotted across the FDR sweep in lines, stratified by simulation mode (color).

Rev3-Q3: Power in detecting fine clusters: The validation data sets the authors considered all have clearly defined clusters, such as those related to pancreas. Can this pipeline be used in distinguishing more subtle cell types, such as various subtypes of CD8 T cells, and naive B, intermediate B, and memory B cells? Understanding these finer scale cell populations is of

concurrent interest, and it would be great if the authors could demonstrate how the proposed pipeline works on these cases.

Rev3-R3: This is a good suggestion; to analyze the potential of our pipeline to distinguish these more subtly divergent populations, we applied our feature selection and clustering approach to a dataset of human PBMCs from 24 donors (12 non-diseased and 12 diabetic). This analysis further highlights the discussion above, in which batch effects can be mitigated through multiplexing, as this dataset was collected using Parse's split-pool combinatorial indexing multiplex approach, rather than performing each replicate separately. This analysis not only showed the powerful sensitivity of the anti-correlation based approach proposed in our manuscript, but also led to the discovery of type-1-diabetes induced increases in relative abundance of subsets of activated classical monocytes, naïve CD4+ T-cells, effector memory CD4+ T-cells, and cytotoxic CD8+ T-cells. While these results required further investigation at the bench, our results point in a promising new direction, show that meaningful subpopulations can be identified even in a single round of clustering. These results are now shown in **Figure 3**.

Figure 3: Application to large real-world datasets. **a**, A heatmap of the top 5 marker genes per cluster are shown for the primary lineages from the full senescent *Tabula Muris* dataset³³, with the last cluster representing a mixture of cell-types from the endocrine pancreas. **b**, When subclustered with anti-correlated feature selection, cell-type droplets (x) as well as classically described leukocyte, α , δ , β , and acinar populations were discovered. Subclustering β cells discovered mixed-lineage droplets with δ and leukocyte cells as well as the rare PP-cell population, but additional subclustering of PP-cells was prevented by anti-correlation-based feature selection. **c**, Selected features for clustering 1-million PBMCs. **d**, Subject-level reverse Percent Maximum Difference (rPMD), shows that Type-1-Diabetes (T1D) subjects are more similar to each other, while control PBMCs are more diverse by cluster composition. **e**, A spring

embedding of a subset of cells from each cluster, color-coded by donor, with sub-plots for T1D and control subjects, showing large-scale uniformity in T1D compared to the heterogeneous control samples. Note that this is for display purposes only, was not used in analysis, and does not represent cell-cell distances, but rather a display of the graph used for clustering. **f**, A heatmap of PMD standardized residuals, which correspond to the significance of how different each subject's relative abundance of all clusters differs from the null expectation of no-difference between subjects. A matching bar-chart shows the T-statistic of cluster level significance for each cluster's differential over-under abundance shown in the heatmap, comparing T1D to controls. Bars are color-coded by significance ($P < 0.05$ after Benjamini-Hochberg). **g**, The spring embedding of the kNN graph is color-coded by significance of differential abundance for each cluster, and additionally color-coded by T1D/control status, then again subset for only the significant clusters.

Reviewer #1 (Remarks to the Author):

The authors have effectively addressed all of my previous concerns, which included extending their analyses to 15 real-world datasets, conducting a more comprehensive benchmarking experiment, comparing their method with state-of-the-art techniques, and exploring a robust range of hyperparameter values. I hope the review process has enhanced the quality of the manuscript.

Comment on the response to Ref #2's comments:

We appreciate the response from the authors and acknowledge their efforts in addressing our concerns. Overall, they have provided clarifications on several key points.

First, they have made it clear that their anti-correlation feature selection method helps determine whether to proceed with further sub-clustering within a cluster, but it does not determine the exact number of clusters for clustering algorithms (as raised in Rev2-Q3). Second, the authors have included a benchmarking experiment with state-of-the-art clustering methods, which is a valuable addition. This will assist readers in selecting the most suitable clustering method when utilizing the proposed feature selection approach for their analysis needs (as mentioned in Rev2-Q4). Moreover, the authors have now explained how their method automatically detects the number of features, eliminating the need for manual hyperparameter tuning, as required by other methods (as addressed in Rev2-Q2).

However, one aspect that remains unclear is how the authors quantify their confidence in the absence of clusters or sub-clusters when zero features are selected (as raised in Rev2-Q1). It would be helpful if the authors could provide an explanation of how they assess the performance of the clustering algorithm in cases where a single cluster is identified. We believe that addressing this point would enhance the understanding of the methodology and its limitations. We appreciate the authors' attention to our concerns and look forward to their response.

Reviewer #3 (Remarks to the Author):

I appreciate the authors' efforts on addressing the three major concerns I have listed in my previous round review.

While the first two concerns have been addressed to a reasonable degree, I do not think the third issue (Power in detecting fine clusters) has been carefully addressed in this revision.

My third concern was whether the proposed method could achieve finer scale cell type differentiation than existing methods. In the response letter, the authors claimed that the approach detected "(T1D) induced increases in relative abundance of subsets of activated classical monocytes, naïve CD4+ T-cells, effector memory CD4+ T-cells, and cytotoxic CD8+ T-cells" in a human PBMC dataset with both control and T1D patients, and that "The results are now shown in Figure 3". In Figure 3, the related panel should be g. However, I cannot see clear evidence of what the authors claimed.

I suggest that the authors demonstrate their claim stated in the response letter in a more clear way in Figure 3, and if necessary, with an additional supplementary figure.

Reviewer #1 (Remarks to the Author):

The authors have effectively addressed all of my previous concerns, which included extending their analyses to 15 real-world datasets, conducting a more comprehensive benchmarking experiment, comparing their method with state-of-the-art techniques, and exploring a robust range of hyperparameter values. I hope the review process has enhanced the quality of the manuscript.

Response to reviewer 1: Thank you for your help in improving our manuscript; we indeed agree that it has significantly improved our work.

Comment on the response to Ref #2's comments:

We appreciate the response from the authors and acknowledge their efforts in addressing our concerns. Overall, they have provided clarifications on several key points.

First, they have made it clear that their anti-correlation feature selection method helps determine whether to proceed with further sub-clustering within a cluster, but it does not determine the exact number of clusters for clustering algorithms (as raised in Rev2-Q3). Second, the authors have included a benchmarking experiment with state-of-the-art clustering methods, which is a valuable addition. This will assist readers in selecting the most suitable clustering method when utilizing the proposed feature selection approach for their analysis needs (as mentioned in Rev2-Q4). Moreover, the authors have now explained how their method automatically detects the number of features, eliminating the need for manual hyperparameter tuning, as required by other methods (as addressed in Rev2-Q2).

Rev2-Q1: However, one aspect that remains unclear is how the authors quantify their confidence in the absence of clusters or sub-clusters when zero features are selected (as raised in Rev2-Q1). It would be helpful if the authors could provide an explanation of how they assess the performance of the clustering algorithm in cases where a single cluster is identified. We believe that addressing this point would enhance the understanding of the methodology and its limitations. We appreciate the authors' attention to our concerns and look forward to their response.

Rev2-R1: When zero features are selected, it is not possible to perform clustering, given that there is no data to perform the clustering upon; we therefore take the result of "no selected genes" as a single cluster result, because clustering is not possible (regardless of the preferred clustering algorithm). We have first, modified the text to provide greater clarity on this, and to improve the readers' understanding of how our method can be pushed to its limits in passing the null-test (ie: single-cluster), we have also added a benchmark (**ED Fig. 3b,c**) that shows a surprising level of robustness, never selecting any features (therefore preventing clustering, yielding only a single group), when the FDR hyperparameter was set to 0.5 or lower. The relevant updates to our manuscript include the following text, benchmark, and figure that directly address the reviewer's question (lines 171-179):

"

Additionally, we sought to clarify which hyperparameters were necessary to pass the null-test. Given the 'detection of a single population' is done indirectly, through returning zero selected features, we benchmarked

the anti-correlation-based feature selection algorithm across these hyperparameters to identify the number of genes selected, and whether the algorithm passes the null-test. We observed that no features were selected (**Extended Data Fig. 3b**) in our null simulations, uniformly passing the null-test (**Extended Data Fig. 3c**), identifying no clusters whenever the FDR parameter was set to values ≤ 0.5 . This result demonstrates strong robustness (good specificity) across hyperparameter space, ensuring that when following our guidance, one will be protected from false discoveries.

”

Extended Data Figure 3: Hyperparameter selection with anti-correlation-based feature selection. a, Scatter plots of 11 different machine learning performance metrics are shown for a hyperparameter sweep of

the FDR and FPR settings when running the anti-correlation algorithm. Three FPR settings were used, and are broken into three vertical panels per metric employed (FPR=0.001 (default setting), 0.01, and 0.1). The x-axis of all plots shows the other parameter (FDR), for which 6 values were used (0.01, 0.1, 0.066 (default), 0.25, 0.5, and 0.99). Plots are color-coded by the simulation mode (SERGIO and Splatter). The means of each metric are plotted across the FDR sweep in lines, stratified by simulation mode (color). **b**, Heatmap showing the average number of selected genes for each combination of the FPR and FDR hyperparameters, n=20 iterations. Red lines indicate recommended hyperparameter ranges. **c**, Heatmap showing the percentage of iterations for which the number of selected genes was 0, therefore indicating a 'passage' of the null test; red lines indicate recommended hyperparameter ranges.

Reviewer #3 (Remarks to the Author):

I appreciate the authors' efforts on addressing the three major concerns I have listed in my previous round review. While the first two concerns have been addressed to a reasonable degree, I do not think the third issue (Power in detecting fine clusters) has been carefully addressed in this revision.

Rev3-Q1: My third concern was whether the proposed method could achieve finer scale cell type differentiation than existing methods. In the response letter, the authors claimed that the approach detected "(T1D) induced increases in relative abundance of subsets of activated classical monocytes, naïve CD4+ T-cells, effector memory CD4+ T-cells, and cytotoxic CD8+ T-cells" in a human PBMC dataset with both control and T1D patients, and that "The results are now shown in Figure 3". In Figure 3, the related panel should be g. However, I cannot see clear evidence of what the authors claimed. I suggest that the authors demonstrate their claim stated in the response letter in a more clear way in Figure 3, and if necessary, with an additional supplementary figure.

Rev3-R1: Many of the details and depth of this analysis were contained within the associated **Extended Data Table 1**, rather than in the primary figure and main text. However, we have now added 5 new extended data figures (89 new panels in total), and a new total of 7 tables within the **Extended Data Table 1a-g** excel file, having re-done much of our analysis more manually, re-merging instances of over-clustering, performing manual marker gene analysis for labeling many of the clusters. We have correspondingly altered the text to reflect this, and have done more to address the reviewer's original request: "Can this pipeline be used in distinguishing more subtle cell types, such as various subtypes of CD8 T cells, and naïve B, intermediate B, and memory B cells?" by hunting through the cluster results for associated marker genes.

Indeed we were able to identify several sub-populations of interest, distinguishing not only the canonical memory and naïve CD4s/CD8s and the B-cell primary lineages, but we were also able to identify continuums of cell-identity within CD4+ memory cells, Tregs, NFKB high B-cells, and populations that were consistent with B1 B-cells, as well as more traditional B2 population. We further found that the T1D over-abundant monocyte population showed a distinct activation pattern relative to its most closely related non-differentially abundant cluster, marked primarily by NAMPT, HIF1A, and FOSB overexpression.

For the interested reader, further exploratory analyses can be performed by downloading the anndata/Scanpy formatted hdf5 file, which holds all annotations in our figshare repository.

The updated text (lines 243-284) and related figures are below:

“

As previously mentioned, several instances of over-clustering occurred, which we manually re-combined where it was determined that UMI depth was the primary difference. Comparison to a previously published large

PBMC dataset³⁵ enabled coarse-grained best-guess cluster labels (**ED Fig. 6**). However, manual analysis was still necessary for accurate labeling.

Among the B-cells, we observed low abundance BLIMP1+/XBP1+ plasma cells (cluster-11)³⁶ (**ED Fig. 7**), that contrasted with CD268+ mature B-cells (clusters-2, -15, -16). Cluster-15 was characterized by up-regulation of NFKB1. Clusters-2 was CD23+, while cluster-16 cells were CD23-, but also contained two regions that were characterized by activated TACI+/CD80+/CD5- and non-activated TACI-/CD80-/CD5+ phenotypes, which may be consistent with B1 (with B1a/B1b subtypes)³⁶. However, definitive identification and further refinement of subsets, is an area of ongoing research³⁷, especially given sensitivity limitations and post-transcriptional regulation in RNA-only assays.

Among T-cells (**ED Fig. 8a,b**), we identified a BLIMP1+ mixed CD4-memory population (cluster-12,21)³⁸, PECAM1+ naïve CD4 (cluster-13,23,25)^{39,40}, and memory and naïve CD8+ T-cells, based on the same markers (clusters-9,6 and -0, respectively) (**ED Fig. 8c,d**), and FOS+/CD97+⁴¹ early activated T-cells, which rapidly downregulate CD4/CD8 after activation⁴² (cluster-17) (**ED Fig. 8f**). Interestingly, a subset of CD4mem cells appeared to show high expression of the TOX exhaustion marker gene⁴³. To further refine this population, which was also moderately increased in T1D (**Fig. 3g**; T-statistic= 2.95, $P=2.2e-2$, BH-corrected t-test; **ED Table 1c-d**), we subset and re-performed feature selection and clustering, successfully identifying 3 discrete populations (**ED Fig. 9a-b**; **ED Table 1g**), with the largest CD4+ effector cluster appeared to exist within a cell-identify marked by a *STAT4/RUNX2-to-TSHZ2/ICOS* expression continuum (**ED Fig. 9c**)^{44, 45}. A remaining CD4+ memory subcluster was *CD25+/FOXP3+/TOX+*, which is consistent with Tregs⁴⁶ (subcluster-7). The final low-abundance cluster (subcluster-11) was characterized by *PECAM1* positivity indicating that these were likely incorrectly co-clustered naïve CD4+ T-cells from the prior clustering-round (**ED Fig. 9**).

In addition to simply characterizing the many different subsets of PBMCs, we further sought to investigate their disease relevance. Notably, we found that the T1D samples were highly similar to each other based on cluster composition, whereas the healthy controls were far more diverse (as measured by reverse Percent Maximum Difference (PMD) (1-PMD, **Figure 3d,e**; **Extended Data Table 1a**). PMD and its standardized residuals quantifies subject-level similarity based on cluster composition robustly to differences in subject level sampling depth⁴⁷.

We found several significantly differentially over-abundant clusters, however, the most differentially abundant was a subset of CD14+/CCR2+/CD115+ classical monocytes (**ED Fig. 10a-c**; T-statistic=10.7, $P=1.34e-25$, BH-corrected t-test; **ED Table 1c-d**)⁴⁸ appearing uniformly over-abundant in T1D subjects (cluster-5,8; **Figure 3f-g**; **Extended Data Table 1**)⁴⁹⁻⁵¹. When comparing this population to its most closely related monocyte population (cluster-10), we found that the T1D over-abundant cluster had significantly higher expression of the activation marker *FOSB*, and functional modulators *NAMPT*⁵² and *HIF1A*, which is induced in inflammation, even in normoxic conditions^{53, 54} (**ED Figure 10d**); differentially expressed genes between these populations are available in **ED Table 1h**. However, further research will be needed to interrogate and independently confirm these findings.

”

New Figures:

Extended Data Figure 6: First draft cluster labels. **a**, A heatmap showing normalized Spearman correlations of the average transcriptomes of clusters discovered here, and in a large reference COVID19 reference dataset. **b**, Spring embedding showing all first-round clusters. **c**, Spring embedding of several lineages of interest and their corresponding clusters.

Extended Data Figure 7: Detailed analysis of B-cells. **a**, Spring embedding of all clusters. **b**, Spring embedding of B-cell clusters. **c**, Expression of canonical B-cell markers overlaid on the spring embeddings, (higher is red, lower expression is grey), and as violin plots. **d**, Expression of plasma cell markers BLIMP1 and XBP1. **e**, Expression of NFKB1, particularly high in the noted cluster-15, as shown in the spring embedding and violin plots. **f**, CD23 expression, particularly high in the noted cluster-16, as shown in the spring embedding and violin plots. **g**, B1-like positivity for TOX, TACI, CD80 and CD5 as shown by expression on spring embeddings and violin plots. Note a small sub-population of CD5+ and CD5- cells within this cluster.

Extended Data Figure 8: Detailed analysis of T-cells. **a**, Spring embedding of all clusters. **b**, Spring embedding of T-cell subsets. **c**, Spring embeddings colored by expression of broad canonical T-cell markers.

d, Expression of TOX and BLIMP1 indicate memory populations. **e**, PECAM1 expression marks naïve populations. **f**, Expression of FOS and CD97 that mark early activation.

Extended Data Figure 9: Detailed analysis of CD4-memory subsets. **a**, Spring embedding of all clusters. **b**, A UMAP projection of the mixed memory CD4 cluster after re-performing feature selection (note that the originally performed spring embedding appeared “knotted” and difficult to see markers of interest due to overlapping populations). **c**, The largest population appeared activated as shown by the plotted known activation and co-stimulatory markers. **d**, CD4+/CD25+/FOXP3+ T-regs also showed TOX positivity. **e**, UMAP showing expression of PECAM1+ indicates that the final sub-cluster-11 was likely contaminant naïve cells.

Extended Data Figure 10: Detailed analysis of differentially abundant classical monocyte population. **a**) Spring embedding of all clusters. **b**, All clusters labeled as classical monocytes. **c**, Expression of canonical markers for monocytes such as CD14, CD16, CCR2, and CD115 as shown with colored spring embeddings and violin plots. **d**, Expression of several of the top differentially expressed genes between the differentially more abundant cluster-5,8 and the non-differentially-abundant cluster-10, as shown on colored spring embeddings and violin plots.

Reviewer #1 (Remarks to the Author):

The authors have appropriately addressed our concern regarding the validation of single clusters found. They have made necessary modifications to the text and included visualizations to elucidate this aspect more effectively.

Reviewer #3 (Remarks to the Author):

The authors have successfully addressed my remaining concerns in the last round review.